# Tidal effects in gravitational waves from neutron stars in scalar-tensor theories of gravity

Gastón Creci[1*], Iris van Gemeren[1†] and Tanja Hinderer[1] Jan Steinhoff[2]

**1** Institute for Theoretical Physics, Utrecht University,
Princetonplein 5, 3584 CC Utrecht, Netherlands, European Union
**2** Max-Planck-Institute for Gravitational Physics (Albert-Einstein-Institute),
Am Mühlenberg 1, 14476 Potsdam-Golm, Germany, European Union

⋆ g.f.crecikeinbaum@uu.nl , † i.r.vangemeren@uu.nl

## Abstract

We compute tidal signatures in the gravitational waves (GWs) from neutron star binary inspirals in scalar-tensor gravity, where the dominant adiabatic even-parity tidal interactions involve three types of Love numbers that depend on the matter equation of state and parameters of the gravitational theory. We calculate the modes of the GW amplitudes and the phase evolution in the time and frequency domain, working up to first order in the post-Newtonian and small finite-size approximations. We also perform several case studies to quantify the dipolar and quadrupolar tidal effects and their parameter dependencies specialized to Gaussian couplings. We show that various tidal contributions enter with different signs and scalings with frequency, which generally leads to smaller net tidal GW imprints than for the same binary system in General Relativity.

# 1 Introduction

Gravitational waves (GWs) are copiously produced by coalescing compact binary systems, with more than ninety confirmed detections to date [1–3] that have already yielded valuable insights into these sources and dynamical spacetime itself. Systems involving neutron stars (NSs) [4–7] are particularly rich in information because they involve strong-field gravity coupled to subatomic matter at supranuclear densities, which remains a longstanding frontier in nuclear physics [8–10]. Extracting this information from GW signals relies on template models used for the detection and parameter estimation analysis. Shortcomings of the templates can thus jeopardize the interpretation of our measurements. This has motivated a significant research efforts to improve the accuracy and physical realism of theoretical models. While theory-agnostic tests of gravity are most commonly used in data analysis, and have already provided constraints on parameterized deviations of the measured waveforms from those in General Relativity (GR) [11–13], complementary theory-specific calculations are needed to connect between empirical measurements and fundamental theory, and to potentially reveal new phenomena to search for.

In this paper, we study GW signatures from tidal effects in NS binary inspirals in scalar-tensor (ST) theories of gravity [14–17]. Such theories involve a scalar field coupled to the metric, which can give rise to *scalarized* NS solutions whose internal structure depends on the scalar field [15, 18–24]. We consider here ST theories in which scalarization of isolated NSs occurs only when their compactness is above a parameter-dependent critical value [15, 19, 20], while black holes and less compact stars are unaltered compared to GR. Thus, ST theories avoid the stringent constraints on deviations from GR imposed by Solar-system experiments [25], but are constrained by observations of binary pulsar, where scalar radiation would lead to accelerated orbital decay relative to GR [19, 26–30]. Complementary constraints come from GW measurements, where scalar radiation and other modifications of the inspiral as well as phenomena such as dynamical scalarization [31], whereby an unscalarized star can become scalarized when reaching a certain separation from the companion, could also play a role [32–39].

The effects of scalarization on a NS's internal structure are encoded in the GW signals, for example, through tidal deformability coefficients that characterize the ratio of the star's induced multipole moments to tidal fields due to the binary companion [40–44]. In ST theories, this induced response for each multipolar order and even parity comprises three different tidal deformabilities [45] in contrast to only one parameter for NSs in GR. This arises because mass and scalar quadrupoles are induced by the gravitational and scalar tidal fields respectively, and moreover, the nonlinear interplay between gravity and matter also results in an induced scalar multipole in response to a tensor tidal field and vice versa. These tidal deformabilities are sensitive to the still poorly constrained equation of state (EoS) of NS matter as well as properties of the scalar field and spacetime [5, 46–50]. While we consider here a specific class of ST theories, our methods have broader applicability to other theories of gravity where scalarized NSs can arise.

There has been much previous work on understanding and modeling the effects of ST gravity on GW signals from binary inspirals. A number of numerical studies of NS binaries in ST theories have analyzed or revealed features due to scalarization, such as the black-hole NS simulations of [51] and those of binary NS mergers in [24, 31, 52, 53]. Analytical waveforms

have been computed in ST theories up to 1.5PN order [18,32,48,54–64,64–67]. The leading
order dipolar tidal effects in the scalar sector have also been modeled [48,68].

In this paper, we complement existing work by assessing the importance of tidal effects up
to quadrupolar order. We compute the different tidal contributions to the Fourier and time-
domain waveform, working to linear order in finite-size effects and to 1PN order in relativistic
effects on the orbital scale. We survey the parameter space of the effects and select several
fiducial binary systems that maximize some of these effects as well as intermediate cases whose
GW signals we analyze in detail, with particular focus on the GW phasing due to the different
kinds of tidal effects.

The paper is organized as follows. In Section 2, we start by discussing GWs in ST theories,
their transformations between different computational frames often used in literature, and the
effective action description of a compact binary system. In Section 3, we derive the leading
tidal effects to 1PN results and compute the waveforms. In Sec. 5, we apply our results to
scalarized NS considering three EoS ranging from stiff to soft and discuss our findings. We
finish with the conclusion and outlook in Sec. 6. We leave the details concerning the PN
calculations and our parameter space study to Appendix B and E, respectively.

The notation and conventions we use are the following. Greek letters $\alpha, \beta, \dots$ denote
spacetime indices, while Latin indices $i, j, \dots$ correspond to spatial components. We use $\nabla_\mu$
to denote the covariant derivative and $\partial_\mu$ for the partial derivative. Capital-letter super and
subscripts, with the exception of the labels T, S and ST, correspond to a string of indices on
a tensor (see e.g. [69] for details), and angular brackets on tensor indices denote the sym-
metric and trace-free (STF) part. For instance, for a unit three-vector $n^i$, the STF tensor
$n_{<L=2>} = n_i n_j - 1/3 \delta_{ij}$, with $\delta_{ij}$ the Kronecker delta. We adopt the Einstein summation
convention on all types of indices, i.e., any repeated indices are summed over. In this work we
set the speed of light equal to unity; $c = 1$.

## 2  Setup and approximation scheme for neutron star binary inspirals in ST theories

### 2.1  Scalar-tensor theories of gravity

Scalar-tensor theories are a class of theories beyond GR that include a scalar field coupled with
the metric. The action for such theories can be formulated in two frames. Historically, it was
first formulated in the so-called Jordan frame,

$$S_{\text{ST}}^{(\text{J})} = \int_{\mathcal{M}} d^4 x \frac{\sqrt{-g_*}}{16\pi G} \left[ F(\phi) R_* - \frac{\omega(\phi)}{\phi} \partial^\mu \phi \partial_\mu \phi - V(\phi) \right] + S_{\text{matter}} \left[ \psi_m, g_{\mu\nu}^* \right], \qquad (1)$$

with $R_*$ the Ricci scalar, $F(\phi)$ an arbitrary function of the scalar field $\phi$, $\omega(\phi)$ a self-interaction
coupling, and $V(\phi)$ its potential. The asterisk is only a notational adornment to distinguish
quantities from those associated to a conformal metric introduced below. In this work, we will
focus on massless scalar fields, so that $V(\phi) = 0$. We note that in (1), the scalar couples only
to the curvature while the matter part is the same as in GR; in particular, there are no effects on
the particle physics of the standard model. However, the scalar coupling to the curvature leads
to complicated equations of motion for the metric fields. Thus, it is convenient to transform to
a different frame where the field equations simplify. Specifically, by performing a conformal
transformation

$$g_{\mu\nu}^* = A(\varphi)^2 g_{\mu\nu}, \qquad (2)$$

with a field-dependent conformal factor that is related to the coupling function by

$$A(\varphi) = \exp\left(-\int d\varphi \frac{F'}{2F\sqrt{\Delta}}\right) , \tag{3}$$

where $\varphi$ is defined by the field redefinition,

$$\frac{d\varphi}{d\phi} = \sqrt{\Delta} , \tag{4a}$$

$$\partial_\alpha \phi = \frac{1}{\sqrt{\Delta}}\partial_\alpha \varphi , \tag{4b}$$

with

$$\Delta \equiv \frac{3}{4}\left(\frac{F'}{F}\right)^2 + \frac{1}{2}\frac{\omega(\phi)}{\phi F} , \tag{5}$$

and a prime denoting a derivative with respect to the argument, $F' = dF/d\phi$, we obtain the action in the Einstein frame,

$$S_{\text{ST}}^{(\text{E})} = \int_{\mathcal{M}} d^4x \frac{\sqrt{-g}}{16\pi G}\left[R - 2g^{\mu\nu}\partial_\mu\varphi\partial_\nu\varphi\right] + S_{\text{matter}}\left[\psi_m, A^2(\varphi)g_{\mu\nu}\right] . \tag{6}$$

In the Einstein frame, the gravitational sector of the action is the Ricci scalar $R$ associated to the metric $g_{\mu\nu}$ with a free scalar field $\varphi$. However, the matter action contains $A(\varphi)$ which couples the metric and scalar field through the matter action. In principle, as the two frames differ only by a conformal transformation, any measurable quantities should be independent of the choice of frame, provided that they are calculated consistently within each frame.

## 2.2 Relevant scales for binary neutron star systems in ST theories

We first consider relevant scales involved in the problem to gain insight into the basic physics dominating different aspects of the binary dynamics and GWs and to define appropriate approximation schemes to model them. Specifically, we consider two non-spinning NSs or a black hole-NS system at large separation $r$ during their early inspiral. Due to the presence of the scalar field in the gravitational theory, depending on the parameters, a scalar configuration inside and around NSs can arise [18]. We assume the binary to move in a quasi-circular motion, slowly descreasing in radius due to the loss of GWs and scalar radiation. Fig. 1 shows a schematic illustration of the systems we consider. The smallest scale that is relevant for modeling the GWs is the size of the bodies in the binary. For a black hole, this is of the order of its Schwarzschild radius $r_S \sim 2GM$, with $M$ its mass, while for NSs, depending on its mass and EoS, it is generally of order $R_{\text{NS}} \sim O(10\text{km})$ equivalent to a few times $r_S$. During the early inspiral, the orbital separation $r$ is much larger than the size of the bodies, $r \gg R_{\text{body}}$. The next larger scale in this setup is the reduced wavelength of the GWs, which is of order $\omega_{\text{GW}}^{-1} \sim O(\omega^{-1})$ with $\omega$ the orbital angular frequency. The largest scale is the distance from the source to the observer denoted by $d$. The above scales are commonly discussed in the literature on PN theory and finite size effects in binary systems in GR, as reviewed e.g. in [70].

In beyond-GR gravity theories, it is often the case that the presence of extra fields introduce new scales into the problem. However, in the massless ST theories we consider here, the scalar configuration (in cases where it exists) is given by a monotonically decreasing function outside the star, in principle extending to infinity. To be able to still assign a characteristic length scale to the scalar configuration we look at the ADM energy of the field

$$E^\varphi = \int_{\mathcal{S}} \sqrt{\gamma}d^3x \, n^\alpha T_{\alpha\beta} t^\beta, \tag{7}$$

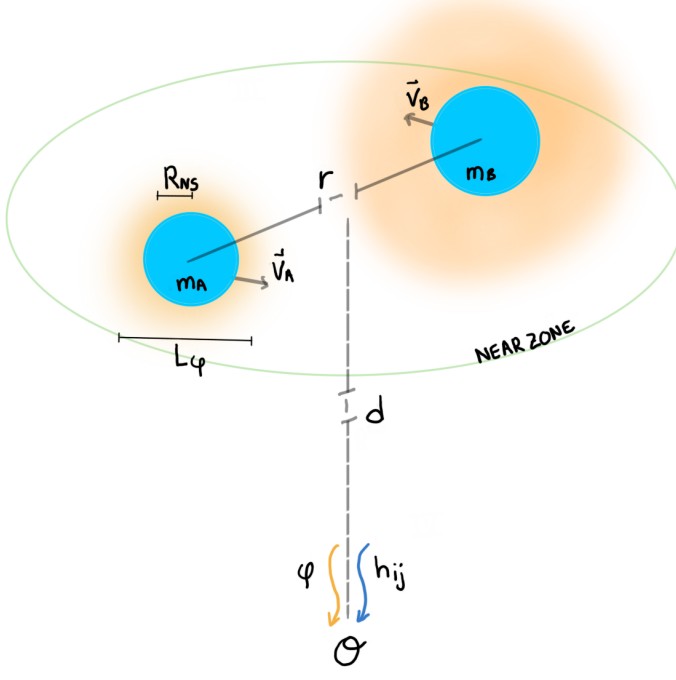

Figure 1: Schematic illustration of the binary systems of NS A and B and their respective masses and velocities. The relevant length scales of the system are labelled.

where $t^\beta$ is the timelike Killing vector $\vec{\partial}/\partial t$, $\mathcal{S}$ the spatial hypersurface with induced metric $\gamma_{\alpha\beta}$ and $n^\alpha$ is the timelike unit normal. The neutron star background metric is obtained by solving the ST field equations with a matter action corresponding to the perfect fluid energy momentum tensor. For details we refer to Sec. 5.1.2 and [45]. The energy momentum tensor of the scalar field will be introduced at the end of this section in (19). Fig. 2 shows the percentage of the scalar field ADM energy as function of the radial distance for an example case NS of $1.4M_\odot$, assuming EoS SLy and ST parameter $\beta_0 = -4.5$. We give a more detailed discussion on the possible parameter choices in Sec. 5. In this case, we find that the ADM energy around the scale of $\sim 5.5R_{NS}$ is around 90% of the total ADM energy at spatial infinity. We repeated this analysis for a range of NS masses and respectively softer to stiffer EoS WFF1, SLy, H4 and for different values of the ST parameter $\beta_0$. For this sweep of the parameter space we found very similar profiles for the ADM energy percentage as shown in Fig. 2 for which 90% of the total ADM energy is always captured in a region smaller than $\mathcal{O}(10)R_{\text{body}}$. Hence despite of the slow falloff of the scalar field itself, its energy is concentrated in a region close to the body. There are no strict criteria for defining the scale of the scalar configuration. However in the overall hierarchy of scales we discuss in this section based on Fig. 2 we can say it is of the order of magnitude of $R_{\text{body}}$. We define therefore the characteristic length scale of the scalar cloud to be

$$L_\varphi \sim 6R_{\text{body}}. \tag{8}$$

Even though the ADM energy related to the scalar field is concentrated close to the compact object, the energy density at the relevant scales during the inspiral can still be significant. To get some insight we did an order of magnitude estimate regarding the density of the field. At a GW frequency around 200Hz during the inspiral part of a coalescing event of a NS binary, their relative distance can be found via $r \sim f\pi/G\alpha M$, with $\alpha$ a ST parameter related to the scalar charges of the NSs, defined in (30), and $M$ the total mass of the binary system. We derive this explicitly in (100). At this radial distance we find the energy density of the scalar

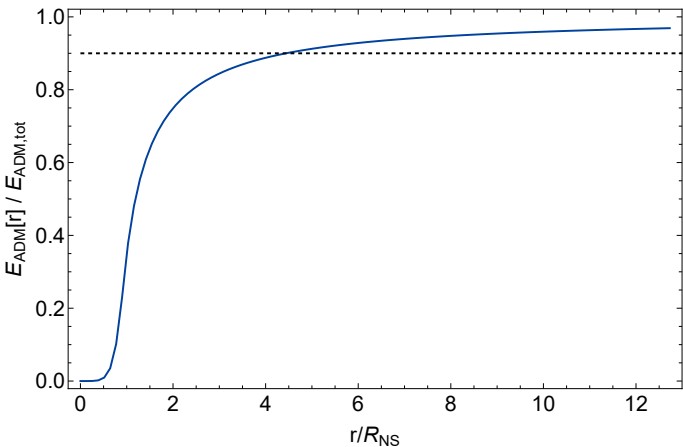

Figure 2: Ratio of the ADM energy as function of the radial distance over the to-tal ADM energy at large distances for a NS of $1.4M_\odot$, considering the SLy EoS and $\beta_0 = -4.5$. The dashed line corresponds to a ratio of 0.9.

field corresponding to the 00 component of the scalar energy momentum tensor (19) to be $\mathcal{O}(10^{11})\text{kg/m}^3$. This is computed for an equal mass NS system of $1.4M_\odot$ with EoS SLy and $\beta_0 = -4.5$. This density surpasses white dwarf like energy densities. For fields with large energy densities, environmental effects like dynamical friction (DF) can leave a significant imprint on the GW signature. The energy loss due to dynamical friction during the inspiral is given by [71,72]

$$\mathcal{F}_{DF} \sim 4\pi\rho \, \eta^2 \frac{G^2 M^2}{v} \ln\Lambda, \tag{9}$$

with $\rho$ the scalar field energy density, $\eta = m_1 m_2/M^2$ the symmetric mass ratio, $v \sim \sqrt{GM/r}$ the velocity of the body moving through the field, which we approximate by the Kelperian velocity. $\ln\Lambda$ denotes the Coulomb algorithm characterized by the impact parameters of the two body encounters. As our order of magnitude is not sensitive to the estimate of $\Lambda$ and it is usually $\mathcal{O}(1)$ [72] we will set it to unity for this estimate. Comparing the DF energy loss to the GW leading order energy flux $\mathcal{F}_{GW} \sim 32/5G^4\eta^2(M/r)^5$ we find the ratio of the fluxes at GW frequency 200Hz to be

$$\frac{\mathcal{F}_{DF}}{\mathcal{F}_{GW}} \sim \mathcal{O}(10^{-1}). \tag{10}$$

Hence the effect of DF on the GWs due to the presence of the scalar field can become quite sig-nificant during the inspiral, moving towards higher energy densities when the radial distance shrinks. We highlight therefore the importance of incorporating environmental effects due to the additional scalar field in future work.

To come back to our discussion on the hierarchy of the length scales in the system as shown in Fig. 1, we divide the problem into different zones. The near zone is defined by $r_{NZ} \ll \omega_{GW}^{-1}$ [73] which captures physics on the orbital scale. The far zone encompasses the region outside the near-zone. In the two zones we use different approximation schemes and match them in the intermediate region. The approximation methods adapted to the differ-ent zones are based on expansions in small dimensionless parameters. Finite size effects are characterized by the ratio between the body scale and the orbital separation

$$\epsilon_{tid} \sim \frac{R_{\text{body}}}{r} \ll 1. \tag{11}$$

As discussed above, for the scalar tidal effects we also work in the approximation that $(L_\varphi/r) \ll 1$. Furthermore the PN weak field and slow motion parameter is

$$\epsilon_{PN} \sim \frac{GM}{r} \sim v^2 \ll 1. \tag{12}$$

We use these parameters to define perturbative expansions that enable us to compute explicit results for the dynamics and GWs of NS binaries in ST theories including tidal effects.

## 2.3   Overview of the approach to compute tidal GW signatures in ST theories

Our calculation in the sections below is structured as follows. We start with a computation of the orbital dynamics on the scale of the relative separation of the two bodies. Because of this large seperation relative to their size, the objects can be described by point particles with small corrections due to their finite size and the scalar condensate. These are described by the introduction of a scalar dependent mass and tidally induced multipole moments. We expand the field equations from action (6) perturbatively to first order in $\epsilon_{PN}$ and $\epsilon_{tid}$ which are small in the near zone. This is followed by a computation of the waveforms at the detector, here one can expand in the large distance to the source $d$, resulting in a multipolar decomposition of the fields. We obtain the waveforms and GW phase in terms of the global charactersitics of the objects. We can relate these parameters to the fundamental properties of the bodies by using the results on the tidal properties for NSs in scalar-tensor theories of [45].

## 2.4   Skeletonized action including tidal effects

When studying the relativistic two-body problem during the early inspiral, we can take advantage of the hierarchy of scales to formulate an effective description valid at the scale of interest. Here, we are interested in the early inspiral where the separation between the bodies $r$ is larger than their characteristic size $R$, $R/r \ll 1$ . To describe the binary system on the large scales of the orbital dynamics and the radiation we make the approximation that the worldtubes of each of the bodies reduce to a fiducial worldline, corresponding to a point-particle at their center of mass, augmented by an infinite sum of multipole moments describing finite-size effects[1]. In this approximation, the total matter action has the form

$$S_{\text{matter}}[g_{\mu\nu}, \varphi, x_A^\mu] = S_{\text{pp}} + S_{\text{tid}}, \tag{13}$$

with

$$S_{\text{pp}} = -\int d\tau \, z \, m(\varphi), \tag{14}$$

the point-particle action with a field-dependent mass $m(\varphi)$, which accounts for the fact that free falling bodies acquire scalar-field dependent terms, therefore violating the strong equivalence principle [56], and $z = \sqrt{-u_\mu u^\mu}$ the redshift factor. For the finite-size effects, we consider here only static tidally-induced scalar and tensor multipole moments of the form

$$Q_L^S = -\lambda_S^\ell E_L^S - \lambda_{ST}^\ell E_L^T, \tag{15a}$$

$$Q_L^T = -\lambda_T^\ell E_L^T - \lambda_{ST}^\ell E_L^S, \tag{15b}$$

where $E_L^{S/T}$ are scalar and tensor tidal fields given by

$$\begin{aligned} E_L^S &= -\nabla_L \varphi, \\ E_L^T &= \frac{1}{z^2} \nabla_{L-2} C_{a_1 \alpha a_2 \beta} u^\alpha u^\beta, \end{aligned} \tag{16}$$

---

[1]Recall that a multipolar series expands in powers of $R/r$ .

where $C_{\mu\alpha\nu\beta}$ is the Weyl tensor, $u^\alpha$ the four-velocity tangent to the worldline, and the indices $L$ are taken to run over $a_1, a_2, \ldots a_\ell$, with $\ell$ the multipolar order. The tidal deformability or Love number coefficients $\lambda^\ell_{S/T/ST}$ characterize the various tidal responses of the star. They contain information from the strong-field regime inside the stars. For the class of ST theories considered in [45], the scalar-tensor Love numbers $\lambda^\ell_{ST}$ were found to be negative, while $\lambda_S$ and $\lambda_T$ were positive. From (15), this implies that the deformation of the star characterized by $\lambda_{ST}$ is of the opposite sign compared to the pure scalar or tensor response.

Using the above definitions, the action for linear, static, even-parity tidal effects is given by [45]

$$
\begin{aligned}
S_{tid} = \sum_\ell \int d\tau \; z \; g^{LP} \times \\
\left( \frac{\lambda^\ell_S}{2\ell!} E^S_L E^S_P + \frac{\lambda^\ell_T}{2\ell!} E^T_L E^T_P + \frac{\lambda^\ell_{ST}}{\ell!} E^T_L E^S_P \right).
\end{aligned}
\tag{17}
$$

## 2.5 Field equations derived from the action

The total action for the system consists of the gravitational action, for which we use the Einstein-frame action (6), together with the skeletonized description of the two bodies (14). Varying with respect to the dynamical fields results in the following field equations

$$
G_{\mu\nu} = 2T^\varphi_{\mu\nu} - 8\pi G(T^{pp}_{\mu\nu} + T^{tid}_{\mu\nu}),
\tag{18}
$$

with

$$
G_{\mu\nu} = R_{\mu\nu} - \tfrac{1}{2}Rg_{\mu\nu},
\tag{19}
$$

$$
T^\varphi_{\mu\nu} = \nabla_\mu\varphi\nabla_\nu\varphi - \tfrac{1}{2}g_{\mu\nu}\nabla_\rho\varphi\nabla^\rho\varphi.
\tag{20}
$$

the Einstein tensor and scalar field energy momentum tensor respectively. Furthermore we define

$$
T^{pp}_{\mu\nu} = \frac{-2}{\sqrt{-g}}\frac{\delta S_{pp}}{\delta g^{\mu\nu}}, \quad T^{tid}_{\mu\nu} = \frac{-2}{\sqrt{-g}}\frac{\delta S_{tid}}{\delta g^{\mu\nu}},
\tag{21}
$$

as the energy momentum tensors associated to the point-particle and tidal action. It is useful to work with the trace-reversed form of the field equations

$$
\begin{aligned}
R_{\mu\nu} - 2\nabla_\mu\varphi\nabla_\nu\varphi + 8\pi G(T^{pp}_{\mu\nu} - \frac{1}{2}g_{\mu\nu}T^{pp}) \\
+ 8\pi G(T^{tid}_{\mu\nu} - \frac{1}{2}g_{\mu\nu}T^{tid}) = 0.
\end{aligned}
\tag{22}
$$

where the trace of the energy momentum tensor is given by $T^{pp} = g^{\alpha\beta}T^{pp}_{\alpha\beta}$ and similarly for the tidal part. The field equation for the scalar field results in

$$
\frac{1}{4\pi G}\Box\varphi = -\frac{1}{\sqrt{-g}}\frac{\delta S_m}{\delta\varphi},
\tag{23}
$$

where $\Box \equiv g^{\alpha\beta}\nabla_\alpha\nabla_\beta$ denotes the d'Alembertian operator.

# 3 Binary dynamics in the PN and tidal approximations

To solve the equations of motion we work perturbatively to linear order in tidal effects and in the post-Newtonian (PN) approximation for relativistic effects to relative 1PN order throughout. The point-particle effects are already known to 1.5 Post-Newtonian (PN) order [18, 63,

281  64, 67]. Tidal effects from dipolar scalar effects have also been computed [68]. As we work in
282  the Einstein frame, our results also resemble those in scalar-Gauss-Bonnet gravity in the zero
283  coupling limit [49, 74, 75]. We do not discuss the known results in detail here and instead
284  focus on the new kinds of tidal contributions at higher multipolar order. We collect the total
285  1PN and tidal corrections of the dynamics, waveforms and phase evolution in Appendix B.

## 3.1 Near zone fields and two body dynamics

287  To construct the tidal contributions to the dynamics of a two body system, we rely on a double
288  perturbative expansion in the PN approximation characterized by the small parameter (12)
289  and in the tidal corrections characterized by (11) and the scalar analogue. We assume each
290  of these corrections to be small and treat them as independent. We focus on the contributions
291  to linear order in the tidal expansion and in the PN approximation. Cross terms of $O(\epsilon_{PN}\epsilon_{\text{tid}})$
292  are discarded as being of higher order in smallness.

293      In the PN approximation, the metric is expanded around flat spacetime and the scalar
294  field around its background value. For convenience, we henceforth use $\epsilon_{PN} \to v^2$ as the PN
295  expansion parameter. To 1PN order, the metric and scalar have the expansion

$$g_{00} = e^{-2U} + \mathcal{O}(v^6), \tag{24a}$$

$$g_{0i} = -4g_i + \mathcal{O}(v^5), \tag{24b}$$

$$g_{ij} = \delta_{ij}e^{2U} + \mathcal{O}(v^4), \tag{24c}$$

$$\varphi = \varphi_0 + \delta\varphi^{(1)} + \mathcal{O}(v^4). \tag{24d}$$

296  From (24d) it also follows that the scalar-dependent mass of body $A$ has the expansion

$$\begin{aligned} m_A(\varphi) = M_A &\left\{1 + q_A\delta\varphi_A^{(1)} + \left(q_A\delta\varphi_A^{(2)} \right.\right. \\ &\left.\left. + \frac{1}{2}\left[(q_A)^2 + \beta_A\right]\delta\varphi_A^{(1)2}\right)\right\} + \mathcal{O}(v^6), \end{aligned} \tag{25a}$$

297  with $M_A$ the ADM mass, $q_A$ the scalar charge, and $\beta_A$ its derivative

$$q_A = \frac{d\,\log m_A(\varphi)}{d\varphi}\bigg|_{\varphi=\varphi_0}, \quad \beta_A = \frac{d\,q_A(\varphi)}{d\varphi}\bigg|_{\varphi=\varphi_0}. \tag{25b}$$

298  The identification of the expansion coefficients in (25a) with the properties of a potentially
299  scalarized NS comes from matching the skeletonized description to the full theory, as discussed
300  in [74]. We adopt the convention from [18] to write the corrections to the mass in terms of the
301  scalar charge instead of the sensitivity parameter $s_A = (d\,\log m_A(\varphi)/d\,\log\varphi)|_{\varphi=\varphi_0}$ that is also
302  often used in the literature [63,65,67]. For the conversion between the charge and sensitivity
303  and other constituent parameters, we refer to Appendix A of [64].

304      Additionally, the fields in (24) are expanded to first order in tidal corrections. However,
305  as was studied in Appendix B of [49], the tidal contributions to the field equations first enter
306  at 1PN order. Here we discard these contributions as higher order, as we focus on the leading
307  Newtonian order tidal corrections. Therefore after substituting the expanded metric (24) and
308  mass (25a) in the field equations (22), (23), and solving the equations for the fields order
309  by order in the PN expansion, we recover the lowest order contributions of the near zone
310  fields [49, 63, 75]

$$U = \frac{GM_A}{r} + (A \leftrightarrow B) + O(v^4),$$

$$\delta\varphi^{(1)} = -\frac{GM_Aq_A}{r} + (A \leftrightarrow B) + O(v^4), \tag{26}$$

where $r$ is the relative separation between the bodies. As these field solutions are obtained in the PN approximation they are only valid in the near zone region introduced in Sec. 2.2. For the tidal corrections we focus on the leading order corrections in a Newtonian order gravitational background, where (16) reduces to

$$
\begin{aligned}
E^S_{L,A} = -\partial_L \varphi &= GM_A q_A \partial_L \left( \frac{1}{r} \right) \\
&= GM_A q_A \frac{(-1)^\ell r^A_{<L>}(2\ell-1)!!}{r^{(2\ell+1)}},
\end{aligned}
\tag{27a}
$$

here $r_{<L>}$ denotes the symmetric trace free (STF) product of $L$ radial terms. For the tensor tidal fields we have

$$
\begin{aligned}
E^T_{L,A} = -\partial_L U &= -GM_A \partial_L \left( \frac{1}{r} \right) \\
&= -GM_A \frac{(-1)^\ell r^A_{<L>}(2\ell-1)!!}{r^{(2\ell+1)}}.
\end{aligned}
\tag{27b}
$$

We note that the scalar charge $q_A$ is negative [45], which makes both kinds of fields in (27) of the same sign for a fixed multipolar order $\ell$. To derive (27), we substituted the near zone fields (26) into the general expressions (16), used the identity (91) for $\ell$ derivatives of $1/r$ and defined $r^{<L>}_A = (x_A - x_B)^{<L>}$, with $x^i_A$ and $x^i_B$ the components of the position vectors of body $A$ and $B$, respectively.

## 3.2 Tidal contributions to the reduced two-body Lagrangian and relative acceleration

To construct the tidal contributions to the binary dynamics, we start by substituting the PN expansions for the mass (25a) and tidal fields (27) in (14) and (17) and obtain the following tidal contributions to the Lagrangian

$$
\frac{dS^A_{tid}}{dt} = G^2 M^2_A \sum_\ell \frac{(2\ell-1)!!}{2r^{2(\ell+1)}} \left( \lambda^\ell_{T,A} + \lambda^\ell_{S,A} q^2_B - 2\lambda^\ell_{ST,A} q_B \right),
\tag{28}
$$

where we used the identity (92) for the contraction of STF unit vectors. Adding the contributions from both bodies yields

$$
L_{tid} = G^2 \mu M \alpha^2 \sum_\ell \frac{(2\ell-1)!!}{2r^{2(\ell+1)}} \zeta_\ell,
\tag{29}
$$

with

$$
\alpha \equiv 1 + q_A q_B,
\tag{30}
$$

$M = M_A + M_B$ the total mass, $\mu = M_A M_B / M$ the reduced mass and a combination of tidal parameters

$$
\zeta_\ell \equiv \frac{M_A}{M_B \alpha^2} \left[ \lambda^\ell_{T,B} + \lambda^\ell_{S,B} q^2_A - 2\lambda^\ell_{ST,B} q_A \right] + (A \leftrightarrow B).
\tag{31}
$$

This tidal contribution adds linearly to the point-mass Lagrangian. The total two-body Lagrangian up to 1PN is given by (95). The expressions for the tidal Lagrangian and acceleration are structurally the same as in GR, where only a tensor deformability appears. Hence, all three Love numbers in ST theories can be taken into account substituting $\zeta^{GR}_\ell \to \zeta^{ST}_\ell$.

From the two-body Lagrangian we compute the relative acceleration from the Euler-Lagrange equations in relative form

$$
\frac{1}{M_A} \frac{\partial L}{\partial x_A} - \frac{1}{M_B} \frac{\partial L}{\partial x_B} = \frac{1}{M_A} \frac{d}{dt} \frac{\partial L}{\partial \mathbf{v_A}} - \frac{1}{M_B} \frac{d}{dt} \frac{\partial L}{\partial \mathbf{v_B}}.
\tag{32}
$$

After transforming to the center-of-mass (CM) frame using the relations in [65] and the identities (93) and (94) for derivatives of $(\partial_L r^{-1})^2$ and contractions of STF multilinears of unit vectors, we obtain

$$a^i_{rel,tid} = -G^2 \alpha^2 M \sum_\ell \frac{(2\ell-1)!!(\ell+1)}{r^{2\ell+3}} n^i \zeta_\ell \,, \tag{33}$$

in agreement with the known results in the limit $\lambda^\ell_S, \lambda^\ell_{ST} \to 0$. The nontidal part of the relative acceleration that adds to (33) to yield the total acceleration can be found in (98).

### 3.3  Tidal contributions to the binding energy

Next, we compute the binding energy of the system from the Lagrangian (95). Assuming a quasi-circular orbit $\dot{r} = \ddot{r} = 0$, we can express the binding energy in terms of the orbital frequency by using $\omega^2 = -\mathbf{a} \cdot \mathbf{r}/r^2$ instead of the coordinate dependent orbital radius parameter. The tidal contribution to this radius-frequency relationship is

$$\omega^2_{tid} = \frac{G\alpha M}{r^3} \left[ \frac{\alpha M}{r} \left( \sum_\ell \frac{(\ell+1)(2\ell-1)!!}{r^{2\ell} M} \zeta_\ell \right) \right]. \tag{34}$$

The full expression including also the point-mass terms is given in (99). Inverting this expression results in (100) with tidal contribution

$$r(x)_{tid} = \frac{G\alpha M}{x} \left[ -\frac{1}{3} x \left( \sum_\ell \frac{(2\ell-1)!!(\ell+1)}{G^{2\ell} \alpha^{2\ell} M^{1+2\ell}} x^{2\ell} \zeta_\ell \right) \right], \tag{35}$$

where we introduce the frequency parameter

$$x = (G\alpha M \omega)^{2/3} \,. \tag{36}$$

Analogously to [49], we then obtain the tidal contribution to the binding energy

$$E_{tid}(x) = -\frac{\mu x}{2} \left[ -\frac{1}{3} \sum_\ell (2\ell-1)!!(4\ell+1) \frac{G^{-2\ell} \zeta_\ell}{M^{2\ell+1} \alpha^{2\ell}} x^{2\ell+1} \right], \tag{37}$$

in agreement with the dipolar expression in [49], and the generic multipolar order result in [76]. The complete result for the binding energy including also the point-mass terms is given in (101).

## 4  Tidal effects in the scalar and gravitational radiation

The gravitational waveforms generated by the dynamics of the binary computed in the previous section can be constructed from the radiative solution of the field equations (18), (23) in the far zone. The field equations can be written as wave equations when introducing the gothic metric $\mathfrak{g}^{ab} = \sqrt{-g} g^{ab}$ [75] and expanding this metric and the scalar field around Minkowski spacetime and the scalar field background respectively. In the harmonic gauge $\partial_\nu \mathfrak{g}^{\mu\nu} = 0$ this results in

$$\begin{aligned}
\Box_\eta h^{\alpha\beta} &= \frac{1}{16\pi G} \mu^{\alpha\beta}, \\
\mu^{\alpha\beta} &= (-g) T^{\alpha\beta}_m + 16\pi G \left( \Lambda^{\alpha\beta} \right) \\
\Lambda^{\alpha\beta} &= 16\pi G (-g) t^{\alpha\beta}_{LL} + h^{\alpha\nu}_{,\mu} h^{\beta\mu}_{,\nu} - h^{\mu\nu} h^{\alpha\beta}_{,\mu\nu},
\end{aligned} \tag{38}$$

where $t_{LL}^{\alpha\beta}$ is the Landau-Lifshitz energy momentum pseudo tensor [77]. The scalar field equation of motion, with $\varphi = \varphi_0 + \delta\varphi$ and $\delta\varphi$ capturing the scalar waveform, is given by

$$\Box_\eta \delta\varphi = 4\pi G \mu_s$$
$$\mu_s = -\frac{1}{\sqrt{-g}}\frac{\delta S_m}{\delta\varphi}. \tag{39}$$

We can write these fields as integrals over the past lightcone by using the retarded Greens function

$$h^{\alpha\beta}(t, \mathbf{x}) = -4G \int d^4x' \frac{\mu^{\alpha\beta}(t', \mathbf{x}')\delta(t' - t + |\mathbf{x} - \mathbf{x}'|)}{|\mathbf{x} - \mathbf{x}'|},$$
$$\delta\varphi(\mathbf{x}) = -G \int d^4x' \frac{\mu_s(t', \mathbf{x}')\delta(t' - t + |\mathbf{x} - \mathbf{x}'|)}{|\mathbf{x} - \mathbf{x}'|}. \tag{40}$$

For the solutions to 1PN order we are considering, the direct integration approach of [73, 78, 79] applies. We split the integration domain of (40) over the past lightcone into the part that lies in the near zone and the part in the far zone. As shown in [63, 73, 75, 78], the far zone contributions are higher than 1PN order. As in the far zone contribution to the integral there are no matter sources, the contribution needs to come from back reaction effects which are generally higher order. For the details on performing these integrals we refer to [63–65, 67, 75]. Below, we discuss the tidal contributions to the waveforms.

## 4.1 Tidal contributions to the scalar waveform

In the near zone we expand (40) in the following multipole expansion based on the fact that far from the source $\mathbf{x}'/|\mathbf{x} - \mathbf{x}'| = \mathbf{x}'/d \ll 1$ to obtain

$$\delta\varphi(\mathbf{x}) = \sum_{l=0}^{\infty} \delta\varphi_\ell(\mathbf{x})$$
$$= -G \sum_{l=0}^{\infty} \frac{(-1)^\ell}{\ell!} \partial_L \left(\frac{1}{d} I_s^L(\tau)\right), \tag{41}$$

where the scalar radiative multipole moments are given by

$$I_s^L = I_s^L \mid_{\text{pp}} + Q_S^L, \tag{42}$$

where the point-particle (pp) contribution is

$$I_s^L(\tau) \mid_{\text{pp}} = \int_{\mathcal{M}} d^3x' \mu_s(\tau, \mathbf{x}') {x'}^L. \tag{43}$$

Here, we introduced the retarded time $\tau = t - d$ and the hypersurface $\mathcal{M}$ cut out by the intersection of the near zone with the constant time hypersurface $t_{\mathcal{M}} = \tau$. As the region $\mathcal{M}$ is bounded, the integral is convergent. For the tidal contribution we use (15a) and (27a), (27b) and convert to the center-of-mass frame using the expressions from [65] to obtain

$$Q_L^S = G \frac{(-1)^\ell r_{<L>}(2\ell - 1)!!}{r^{2\ell+1}} \bar{\zeta}_\ell. \tag{44}$$

here we redefined $r_{<L>} = r_{<L>}^A$ and the two body dependencies are captured in the tidal coefficient

$$\bar{\zeta}_\ell = -M_A q_A \lambda_{S,B}^\ell - M_B q_B (-1)^\ell \lambda_{S,A}^\ell + M_A \lambda_{ST,B}^\ell + M_B (-1)^\ell \lambda_{ST,A}^\ell, \tag{45}$$

the factor $(-1)^\ell$ arises from $r^B_{<L>} = (-1)^\ell r^A_{<L>}$. We are interested in radiation near future null infinity, where $d$ is very large. Thus, when derivatives in (41) act on $1/d$ they will give strongly suppressed contributions which we neglect. To compute derivatives of the multipole moments we use that

$$\partial_i I_s(\tau) = \frac{\partial \tau}{\partial \mathrm{x}^i} \frac{dI_s}{d\tau} = -N^i \frac{dI_s}{d\tau} = -N^i \frac{dI_s}{dt}, \tag{46}$$

where $N = (\mathbf{x} - \mathbf{x}')/d$ is a unit vector pointing from the source to the field point. Thus, the multipolar scalar waves near null infinity are given by

$$\delta\varphi_\ell(\mathrm{x}) = \frac{G}{d} \frac{N_L}{\ell!} \left(\frac{\partial}{\partial t}\right)^\ell I_s^L + \mathcal{O}\left(d^{-2}\right). \tag{47}$$

From (42) it follows that the scalar waveform consist of the point-particle and linear tidal contributions. We express the latter more explicitly by rewriting time derivatives of the tidal multipoles (44) using the generalized Leibniz rule

$$\left(\frac{\partial}{\partial t}\right)^\ell r^{-(2\ell+1)} r_{<L>} = \sum_{k=0}^{\ell} \binom{\ell}{k} \partial_t^{\ell-k} r_{<L>} \partial_t^k r^{-(2\ell+1)}. \tag{48}$$

This can be further manipulated using Faà di Bruno's formula [80]

$$\left(\frac{\partial}{\partial t}\right)^\ell r^{-(2\ell+1)} r_{<L>} = \sum_{k=0}^{\ell} \frac{\ell!}{k!(\ell-k)!} \partial_t^{\ell-k} r_{<L>} \sum_{p=1}^{k} \frac{(2\ell+p)!(-1)^p}{(2\ell)! r^{2\ell+p+1}} B_{k,p}(\dot r, \ddot r, \ldots, r^{k-p+1}), \tag{49}$$

with $B_{k,p}$ the incomplete Bell polynomials. Using this, the total tidal contribution to the scalar waveform can be written as

$$\delta\varphi_{tid} = \sum_{\ell}\sum_{k=0}^{\ell}\sum_{p=1}^{k} \frac{G^2 N^L \bar\zeta_\ell}{d} \frac{(2\ell+p)!(-1)^{p+\ell}}{k!(\ell-k)!2^\ell} \frac{\partial_t^{\ell-k} r_{<L>}}{r^{2\ell+p+1}} B_{k,p}(\dot r, \ddot r, \ldots, r^{k-p+1}). \tag{50}$$

For the total 1PN scalar waveform with tidal corrections see (101).

## 4.2 Tidal contributions to the tensor waveform

Similar to the scalar waveform, the near-zone contributions to the tensor radiation fields can be expressed as a multipole expansion, though some differences arise because the source term of the field equation is a tensor, as explained in Sec. C of [78]. This leads to the radiative fields

$$h^{ij}(t, \mathbf{x}) = \frac{2G}{d} \sum_{\ell=0}^{\infty} \frac{1}{\ell!} N_{L-2} \left(\frac{\partial}{\partial t}\right)^\ell I^{ijL-2} + O\left(d^{-2}\right), \tag{51}$$

where $I^{ijL-2}$ are the tensor radiative multipole moments given by

$$I^{ijL-2} = I^{ijL-2}\big|_{\text{EW}} + Q_T^{ijL-2}. \tag{52}$$

Here, the first term are related to the Epstein-Wagoner multipole moments for point masses [81] and the second one the tidal multipoles (15b). From (15b) and (27a), (27b) we obtain, after converting to the center-of-mass frame using the expressions from [65] and proceeding as in (50),

$$Q_L^T = G \frac{(-1)^\ell r_{<L>}(2\ell-1)!!}{r^{2\ell+1}} \tilde\zeta_\ell, \tag{53}$$

with

$$\tilde\zeta_\ell = -M_A q_A \lambda_{ST,B}^\ell - M_B q_B (-1)^\ell \lambda_{ST,A}^\ell + M_A \lambda_{T,B}^\ell + M_B (-1)^\ell \lambda_{T,A}^\ell. \tag{54}$$

Using (48), (49) we obtain for the direct tidal contribution to the tensor waveform

$$h_{Q^T}^{ij} = \sum_{\ell=2}^{} \sum_{k=0}^{\ell} \sum_{p=1}^{k} \frac{4G^2 N^{L-2} \tilde{\zeta}_\ell}{d} \frac{(2\ell+p)!(-1)^{p+\ell}}{k!(\ell-k)!2^\ell} \frac{\partial_t^{\ell-k} r_{<L>}}{r^{2\ell+p+1}} B_{k,p}(\dot{r}, \ddot{r}, \ldots, r^{k-p+1}). \tag{55}$$

Additionally, the time derivatives in (51) acting on the point mass multipoles in (52) introduce tidal contributions coming from the relative acceleration (33). These involve terms proportional to the parameter $\zeta_\ell$ defined in (31). Together with this contribution, the tidal terms in the tensor waveform read

$$h_{tid}^{ij} = -\sum_\ell \frac{4G^3 \mu \alpha^2 M^2 (1+\ell)(2\ell-1)!! \zeta_\ell}{d\, r^{2(2+\ell)}} r^i r^j + \sum_{\ell=2}^{} \sum_{k=0}^{\ell} \sum_{p=1}^{k} \frac{4G^2 N^{L-2} \tilde{\zeta}_\ell}{d} \times$$
$$\frac{(2\ell+p)!(-1)^{p+\ell}}{k!(\ell-k)!2^\ell} \frac{\partial_t^{\ell-k} r_{<L>}}{r^{2\ell+p+1}} B_{k,p}(\dot{r}, \ddot{r}, \ldots, r^{k-p+1}). \tag{56}$$

For the full 1PN tensor waveform see (103).

Using the standard convention [78] of an orthonormal triad composed of the vectors **N** in the radial direction of the observer, $\hat{\mathbf{p}}$ along the intersection of the orbital plane with the sky and $\hat{\mathbf{q}} = \mathbf{N} \times \hat{\mathbf{p}}$, the plus and cross polarizations of the waveform are given by

$$h_+ = \frac{1}{2} \left( \hat{p}_i \hat{p}_j - \hat{q}_i \hat{q}_j \right) h^{ij},$$
$$h_\times = \frac{1}{2} \left( \hat{p}_i \hat{q}_j + \hat{q}_i \hat{p}_j \right) h^{ij}. \tag{57}$$

For contracting the vectors, we use the identities given in [78] but do not write out the explicit results here.

## 4.3 Gravitational waves in the Jordan frame

The scalar and gravitational radiation computed in Sec. 4 were based on the Einstein-frame formulation of the ST theory, as discussed in Sec. 2.5. Here, we review the transformation of these results to the Jordan frame. The fact that the matter sector of the Jordan-frame formulation of ST theories is the standard model of particle physics implies that one can use existing results for how a GW detector measures signals in that frame. In Appendix D we provide the derivation of the results for the transformations and further discussion based on geodesic deviation and the frame transformation (2) and (4), generalizing the results of [63, 82, 83] to generic coupling functions. We obtain that the Jordan-frame waveform is given in terms of Einstein-frame quantities by

$$h_{ij}^{\text{Jordan}} \simeq h_{ij} + \frac{2A'(\varphi_\infty)}{A(\varphi_\infty)} \delta\varphi\, \delta_{ij}, \tag{58}$$

where $h_{ij}$ is the waveform computed in Sec. 4.2 and given explicitly by (103), and $\delta\varphi$ the scalar waveform computed in Sec. 4.1 and given explicitly in (101). The approximation is that we work in a region of spacetime asymptotically far from the binary source, where the scalar field is dominated by its constant asymptotic value $\varphi_\infty$, and to linear order in small perturbations around an asymptotic background, both for the metric and scalar fields. Applying the decomposition into the tensorial plus and cross polarizations (57) to (58) and using the orthonormality of the spatial triad shows that the tensor polarization amplitudes in the Jordan frame $h_{+/\times}^{\text{Jordan}} = h_{+/\times}$ coincide with those in the Einstein frame given in (57). In addition, the scalar contribution to the GWs in (58) gives rise to an extra scalar polarization component of the GWs. Depending on the coupling function and cosmological value of the scalar field, this contribution may, however, be suppressed by many orders of magnitude.

### 4.4 Energy fluxes

The gravitational and scalar radiation cause an energy flux out of the binary system proportional to the angular integral over the square of the time derivative of the waveforms. Similar to Sec. 3.3 we are interested in the fluxes and phase evolution in terms of the coordinate independent orbital frequency for quasi circular orbits. We substitute (100) in the waveforms (101), (103) and expand perturbatively to linear order in the tidal contributions. This requires explicitly evaluating $r_{<L>}$ in (50), (56). Hence from this section onward we consider only multipole moments up to $\ell = 2$, where all three different tidal contributions (S/T/ST) appear. Below we discuss these tidal contributions to the scalar and tensor energy fluxes.

#### 4.4.1 Tidal contributions to the scalar energy flux

The scalar energy flux can be obtained from the scalar waveform via the surface integral

$$\mathcal{F}_S = \frac{d^2}{4\pi G} \oint \delta\dot{\varphi}^2 \, d^2\Omega. \tag{59}$$

We perform the angular integral over the products of unit vectors $N_i$ using the identities from [69] and substitute (101) to obtain

$$\mathcal{F}_{S,tid} = \frac{4G^3 M S_- \alpha^{3/2} \mu}{3r^6} \left( 9\dot{r} - 3v^2 + \frac{2GM\alpha}{r} \right) \bar{\zeta}_1 + \frac{8G^4 M^2 S_-^2 \alpha^4 \mu^2}{3r^7} \left( 2\zeta_1 + 9\frac{\zeta_2}{r^2} \right), \tag{60}$$

with

$$S_\pm \equiv \frac{q_A \pm q_B}{2\sqrt{\alpha}}, \tag{61}$$

and $\alpha$, $\zeta_\ell$ and $\bar{\zeta}_\ell$ defined in (30), (31) and (45) respectively. Assuming quasi-circular orbits, substituting (100), and expressing the flux in terms of the frequency parameter $x$ defined in (36), we obtain

$$\mathcal{F}_{S,tid} = x^7 \left( -\frac{4S_- \mu \bar{\zeta}_1}{3\alpha^{9/2} G^3 M^5} + \frac{16S_-^2 \mu^2 \zeta_1}{9\alpha^3 G^3 M^5} \right) + x^9 \frac{S_-^2 \mu^2 \zeta_2}{\alpha^5 G^5 M^7}. \tag{62}$$

In the expressions above two types of tidal contributions arise. The $\bar{\zeta}_1$ term results from (50) due to the induced tidal moments. We find that only the odd $\ell$ give a nontrivial contribution to the integral. The $\zeta_{\ell=1,2}$ terms come in via the relative acceleration contributions (33) which arise from the time derivative to the waveform. The full 1PN scalar flux for the point-mass part is given by (104) and (105).

#### 4.4.2 Tidal contribution to the tensor energy flux

The tensor energy flux is constructed via

$$\mathcal{F}_T = \frac{d^2}{32\pi G} \oint \dot{h}_{\mathrm{TT}}^{ij} \dot{h}_{\mathrm{TT}}^{ij} d^2\Omega, \tag{63}$$

where TT denotes the transverse-traceless piece obtained with the aid of the projection operator

$$P^{ij} = \delta^{ij} - N^i N^j \tag{64}$$

to reduce to the transverse part and removing all traces. Note that the derivatives to the tensor waveforms are taken in the $TT$ gauge[2]. The angular integral is again computed using the

---

[2]Using the definition of the projector operator one can use the identity $\left( P^{ik}P^{jl} - \frac{1}{2}P^{ij}P^{kl} \right)\left( P^{im}P^{jn} - \frac{1}{2}P^{ij}P^{mn} \right) = P^{km}P^{ln} - \frac{1}{2}P^{kl}P^{mn}$ to simplify the integral (63) to $\mathcal{F}_T = \frac{\mu^2}{32\pi G} \oint \left( 4\dot{Q}^{ij}\dot{Q}^{ij} - 8N^{ln}\dot{Q}^{kl}\dot{Q}^{kn} + 2N^{klmn}\dot{Q}^{kl}\dot{Q}^{mn} \right) d^2\Omega$. Here $Q$ is defined as $Q_{ij} = \tilde{Q}_{ij} - \frac{1}{3}\delta_{ij}\tilde{Q}_k^k$ with $\tilde{Q}_{ij} = \frac{d}{2G\mu}h_{ij}$.

467  identities in [69]. Substituting the explicit expression for the tensor waveform (103) into (63)
468  we find the following expression for the tidal contributions to the tensor flux

$$\mathcal{F}_{T,tid} = \frac{48G^3 M \alpha \mu}{5r^8} \left( 100\dot{r}^4 - 105\dot{r}^2 v^2 + 15v^4 + 18\frac{GM\alpha}{r}\dot{r}^2 - 11\frac{GM\alpha}{r}v^2 \right) \tilde{\zeta}_2$$
$$- \frac{64G^4 M^2 \alpha^3 \mu^2}{15r^7}(7\dot{r}^2 - 6v^2)\zeta_1 - \frac{192G^4 M^2 \alpha^3 \mu^2}{5r^9}(4\dot{r}^2 - 3v^2)\zeta_2 \tag{65}$$

469  with $\zeta_\ell$ and $\tilde{\zeta}_\ell$ defined in (31) and (54). Again assuming quasi-circular orbits and expressing
470  the flux in terms of the frequency, we obtain

$$\mathcal{F}_{T,tid} = x^8 \frac{256\mu^2 \zeta_1}{15\alpha^4 G^3 M^5} + x^{10}\left( \frac{192\mu\tilde{\zeta}_2}{5\alpha^7 G^5 M^7} + \frac{384\mu^2 \zeta_2}{5\alpha^6 G^5 M^7} \right). \tag{66}$$

471  Similar to the scalar flux we find again two types of tidal contributions. The $\tilde{\zeta}_2$ term results
472  from (56) due to the induced tidal moments. Now only the even $\ell$ contribute to the integral.
473  The $\zeta_{\ell=1,2}$ terms come in via the relative acceleration. For the 1PN tensor flux see (107) and
474  (108). Together with the scalar flux in (104) the total flux is the sum of the two contributions
475  $\mathcal{F} = \mathcal{F}_S + \mathcal{F}_T$.

## 4.5 GW phase evolution

477  During the quasi-circular inspiral, the corrections to the radiation compared to the radiation in
478  GR are accumulated over many GW cycles in the phase evolution. In this regime, the motion is
479  approximately adiabatic, with $\dot{\omega}/\omega^2 \ll 1$ and there is an energy balance between the binding
480  energy of the system and the radiative energy flux $\dot{E}(\omega) = -\mathcal{F}(\omega)$. Together with the change
481  in orbital phase $\dot{\phi} = \omega$ this gives the system of differential equations

$$\frac{d\phi}{dt} - \omega = 0, \quad \frac{d\omega}{dt} + \frac{\mathcal{F}(\omega)}{E'(\omega)} = 0. \tag{67}$$

482  As the results for $E(x)$ and $\mathcal{F}(x)$ are only available perturbatively, there are different choices
483  for how to solve (67), as reviewed e.g. in [84]. With regard to data analysis it is useful to
484  obtain the expressions for the phase evolution in the Fourier domain. The Fourier transform
485  of the gravitational strain measured by a detector and denoted by $\tilde{h}$ can be obtained using the
486  stationary phase approximation (SPA) [85]

$$\tilde{h}^{\mathrm{SPA}}(f) = \mathcal{A}\sqrt{\frac{2\pi}{M\dot{\omega}}}e^{-i[\psi(f)+\pi/4]}, \tag{68}$$

487  with $\mathcal{A}$ the amplitude and $\psi \equiv 2\phi(t(f)) - 2\pi f t(f)$ the Fourier phase. One can rewrite (67)
488  as a second order differential equation in terms of the Fourier phase

$$\frac{d^2\psi(\omega)}{d\omega^2} = -2\frac{E'(\omega)}{\mathcal{F}(\omega)}. \tag{69}$$

489  Using (36) this leads to

$$\psi = -\int \left( \int \frac{E'(x)}{\mathcal{F}(x)}dx \right) \frac{3\sqrt{x}}{G\alpha M}dx + \phi_c - 2\pi f t_c, \tag{70}$$

490  where $\phi_c$, $t_c$ are integration constants determined by the choice of reference point in the
491  evolution. To solve (70) we use the Taylor F2 approximant [84], where the integrand is
492  expanded perturbatively in $x$ and in tidal corrections using the explicit expressions for the

493 fluxes (105), (108) and binding energy (101). The presence of both scalar and gravitational
494 radiation in the system leads to different behaviors depending on the frequency regime: for
495 small frequencies, the scalar dipole terms in the total flux (proportional to $x^4$ in (105)) are
496 dominant, while for larger frequencies the tensor quadrupole terms (proportional to $x^5$ in
497 (108)) become the dominant contributions. These regimes are commonly referred to as the
498 dipolar driven (DD) and quadrupolar driven (QD) regimes respectively. Specifically, based on
499 the leading order contributions to the flux, the regime for which the scalar dipole dominates
500 is

$$x^{\mathrm{DD}} \ll \frac{5 \mathcal{S}_-^2 \alpha}{24} \text{ or } f^{\mathrm{DD}} \ll \left(\frac{5}{24}\right)^{3/2} \frac{\mathcal{S}_-^3 \sqrt{\alpha}}{\pi G M}, \tag{71}$$

501 see [49] for a study of the validity of this approximation to the transition frequency between
502 the two regimes.

### 4.5.1 Phase evolution in the Dipolar-Driven domain

504 For frequencies in the regime denoted by (71) the dipolar term in the total flux is leading. We
505 factor out this term and expand the ratio up to 1PN, which gives the following form

$$\frac{E'(x)}{\mathcal{F}^{\mathrm{DD}}(x)} = \frac{-3\alpha G M}{8\eta S_-^2 x^4} \left[1 + \left(E_0' - f_2^{\mathrm{DD}}\right)x\right], \tag{72}$$

506 with the symmetric mass ratio

$$\eta = \frac{\mu}{M} \tag{73}$$

507 and the coefficient $f_2^{DD}$ given by (110). Substituting (72) in (70) gives the following result for
508 the Fourier phase angle in the DD domain (111), where we show only the tidal contributions
509 here

$$\psi_{tid}^{DD} = \frac{1}{4\eta \mathcal{S}_-^2 x^{3/2}} \left\{ x^3 \left[ \rho_{\mathrm{tid}}^{\mathrm{DD}} \log(x) - \frac{2}{3} \rho_{\mathrm{tid}}^{\mathrm{DD}} \right] \right.$$
$$\left. - \frac{270\zeta_2}{7\alpha^4 G^4 M^5} x^5 \right\} + \phi_c - 2\pi f t_c, \tag{74}$$

510 with

$$\rho_{\mathrm{tid}}^{\mathrm{DD}} = \frac{3}{G^2\alpha^2 M^3 \eta} \left( -\frac{\bar{\zeta}_1}{\alpha^{3/2} M S_-} + 16\eta\zeta_1 \right). \tag{75}$$

511 The total phase including also the point-mass terms is given in (111). Looking at the prefactor
512 of (74) we see that the phase angle diverges for $S_- \to 0$, where $S_-$ was defined in (61). This
513 parameter vanishes for an equal mass system where both bodies have the same scalar charge $q$.
514 However, as the scalar flux terms are proportional to $S_-$ the scalar radiation, and therefore the
515 DD regime, vanishes for these systems and the expansion used to obtain $\psi_{DD}$ which assumed
516 the scalar flux is leading is invalid and would need to be modified.

### 4.5.2 Phase evolution in the Quadrupolar-Driven domain

518 For frequencies above (71) the quadrupolar contribution in the total flux is leading. We split
519 the total flux in its scalar and tensor contributions and factor the dipolar and quadrupolar term
520 labelling them non-dipolar and dipolar respectively. This leads to $\mathcal{F} = \mathcal{F}_{\mathrm{non\text{-}dip}} + \mathcal{F}_{\mathrm{dip}}$ with

$$\mathcal{F}_{\mathrm{non\text{-}dip}} = \frac{32\eta^2\xi}{5G\alpha^2} x^5 \left(1 + f_2^{nd} x\right),$$
$$\mathcal{F}_{\mathrm{dip}} = \frac{4\mathcal{S}_-^2\eta^2}{3G\alpha} x^4 \left(1 + f_2^d x\right), \tag{76}$$

where

$$\xi = 1 + \mathcal{S}_+^2 \alpha / 6, \tag{77}$$

and the other coefficients given in (115). We then expand the ratio in (70) as

$$\frac{E'(x)}{\mathcal{F}(x)} \simeq \frac{E'(x)}{\mathcal{F}_{\text{non-dip}}(x)} \left( 1 - \frac{\mathcal{F}_{\text{dip}}(x)}{\mathcal{F}_{\text{non-dip}}(x)} \right), \tag{78}$$

which results in

$$\frac{E'(x)}{\mathcal{F}(x)} \simeq -\frac{5GM\alpha^2}{64\eta\xi x^5} \left[ 1 + \left( E'_0 - f_2^{nd} \right) x \right] + \frac{25GM\alpha^3 \mathcal{S}_-^2}{1536\xi^2 \eta x^5} \left[ 1 + \left( E'_0 - 2f_2^{nd} + f^d \right) x \right]. \tag{79}$$

Substituting (79) in (70) gives the Fourier phase in the QD domain (116), with the tidal contributions

$$\psi^{\text{QD}} = \psi_{\text{non-dip}} + \psi_{\text{dip}} + \phi_c + 2\pi f t_c, \tag{80}$$

with

$$\psi_{\text{non-dip,tid}} = \frac{3\alpha}{128\eta\xi x^{5/2}} \left[ \rho_{\text{tid}}^{nd,1} x^3 + \rho_{\text{tid}}^{nd,2} x^5 \right], \tag{81a}$$

$$\psi_{\text{dip,tid}} = -\frac{5\mathcal{S}_-^2 \alpha^2}{1792\eta\xi^2 x^{7/2}} \left[ \rho_{\text{tid}}^{d,1} x^3 + \rho_{\text{tid}}^{d,2} x^5 \log(x) - \frac{2}{3} \rho_{\text{tid}}^{d,2} x^5 \right], \tag{81b}$$

The non-dipolar tidal coefficients are given by

$$\rho_{\text{tid}}^{nd,1} = \frac{400\zeta_1}{3\alpha^2 G^2 M^3} + \frac{160\zeta_1}{3\alpha^2 G^2 M^3 \xi}, \tag{81c}$$

$$\rho_{\text{tid}}^{nd,2} = -\frac{24\tilde{\zeta}_2}{\alpha^5 G^4 M^6 \xi \eta} - \frac{216\zeta_2}{\alpha^4 G^4 M^5} - \frac{48\zeta_2}{\alpha^4 G^4 M^5 \xi}, \tag{81d}$$

and the dipolar parts are

$$\rho_{\text{tid}}^{d,1} = -\frac{35\bar{\zeta}_1}{2\alpha^{7/2} G^2 M^4 S_- \eta} - \frac{280\zeta_1}{3\alpha^2 G^2 M^3} - \frac{280\zeta_1}{3\alpha^2 G^2 M^3 \xi}, \tag{81e}$$

$$\rho_{\text{tid}}^{d,2} = -\frac{140\tilde{\zeta}_2}{\alpha^5 G^4 M^6 \eta\xi} - \frac{560\zeta_2}{\alpha^4 G^4 M^5} - \frac{280\zeta_2}{\alpha^4 G^4 M^5 \xi}. \tag{81f}$$

### 4.5.3 Expressions for the tidal phasing in ready-to-use form

The tidal phase contributions (81) can be expressed as

$$\begin{aligned} \psi_{\text{tid}}^{\text{QD}} = \quad & \frac{3}{128\eta x^{5/2}} \Big[ c_2 S_- x^2 + c_3 x^3 \\ & + c_4 S_-^2 \left( \log x - \frac{2}{3} \right) x^4 + \left( \frac{39}{2\alpha^5 \xi^2} \tilde{\Lambda} + c_5 \right) x^5 \Big], \end{aligned} \tag{82}$$

where the various coefficients $c_i$ for $i = 2, 3, 4, 5$ are given explicitly in terms of masses, scalar charges, and tidal deformabilities of the bodies in (120)– (123) in appendix C.1 and $\alpha$ and $\xi$ were defined in (30) and (77), with $S_\pm$ given in (61) and $x$ defined in (36), which includes a dependence on $\alpha$. We have also introduced the parameter $\tilde{\Lambda}$ having the same functional form as in GR and involving the quadrupolar tensor deformabilities [42, 86]

$$\tilde{\Lambda} = \frac{16}{13M^5 G^4} \left[ (11M_B + M) \frac{\lambda_A^T}{M_A} + (11M_A + M) \frac{\lambda_B^T}{M_B} \right], \tag{83}$$

however, as $\lambda^T$ are computed within ST gravity, $\tilde{\Lambda}$ may differ from its value in GR.

The coefficients $c_2$ and $c_3$ in (82) depend only on scalar dipolar tidal effects, while $c_4$ and $c_5$ involve all quadrupolar tidal parameters. Moreover, for the case of identical NSs, $S_-$ defined in (61) vanishes and hence the effects encapsulated in $c_2$ and $c_4$ do not contribute in that case. The remaining coefficients for identical masses $m$, scalar charges $q$, and tidal deformabilities $\lambda_\ell$ are given by

$$c_3 \mid_{A=B} \;=\; \frac{1680q^2(1+\frac{5}{42}q^2)}{(1+q^2)^5(6+q^2)^2}\Lambda_1^S, \tag{84}$$

$$c_5 \mid_{A=B} \;=\; \frac{702}{(1+q^2)^5(6+q^2)^2}\left[\frac{7}{26}q^2\Lambda_2^T\right.$$
$$\left. +\frac{q\left(26+5q^2\right)}{26}\Lambda_2^{ST} - \frac{q(22+3q^2)}{26}\Lambda_2^S\right]. \tag{85}$$

Here, we have defined the dimensionless deformabilities

$$\Lambda_\ell = \frac{\lambda_\ell}{G^{2\ell}m^{2\ell+1}}. \tag{86}$$

We also note that for the case of identical NSs, $\tilde{\Lambda} = \Lambda_2^T$ and $\alpha^5\xi^2 = (1+q^2)^5(6+q^2)^2/36$. Finally, we see that in the case of GR, where $S_- = c_3 = c_5 = 0$ and $\alpha = \xi = 1$ we recover the standard result for the leading-order quadrupolar tidal effects [42, 86].

# 5 Case study of neutron star binaries

## 5.1 Set-up and properties of single neutron stars

We next apply the general results of the previous section to NS binary and BH-NS systems in specific classes of ST theories.

### 5.1.1 Choice of coupling

We consider ST theories characterized by the Damour-Esposito-Farèse coupling [20],

$$A(\varphi) = e^{\frac{1}{2}\beta_0\varphi^2} . \tag{87}$$

These theories contain two arbitrary parameters; the above entered $\beta_0$ and $\varphi_\infty$, the value of the scalar field at spatial infinity. One can rewrite these free quantities by the following parameters

$$\alpha_0 = \alpha(\varphi_\infty) = \partial \ln A(\varphi_\infty)/\partial \varphi_\infty = \beta_0\varphi_\infty, \tag{88}$$

capturing the strength of the coupling between the scalar field to matter in the asymptotic limit and it's derivative

$$\beta_0 = \partial \alpha(\varphi_\infty))/\partial \varphi_\infty, \tag{89}$$

as solar system test probe directly $\alpha_0$ or combined parameters of $\alpha_0$ and $\beta_0$ [19,20,28]. When $\beta_0$ is sufficiently negative, $\beta_0 \lesssim -4.3$ [22], NS solutions become scalarized. That is, in these cases NSs are not only described by the mass and EoS relating the pressure and density, but also by the properties of the scalar field. This threshold value of $\beta_0$ is largely insensitive to the choice of $\alpha_0$, which for increasing $|\alpha_0|$ mostly smooths the sudden scalarization effect [15,24], leading to scalarized NSs over the whole mass range. For our case studies we adopt the choice of a cosmological background scalar field $\varphi_\infty = 10^{-3}$ and $\beta_0 = (-4.5, -6)$ corresponding to

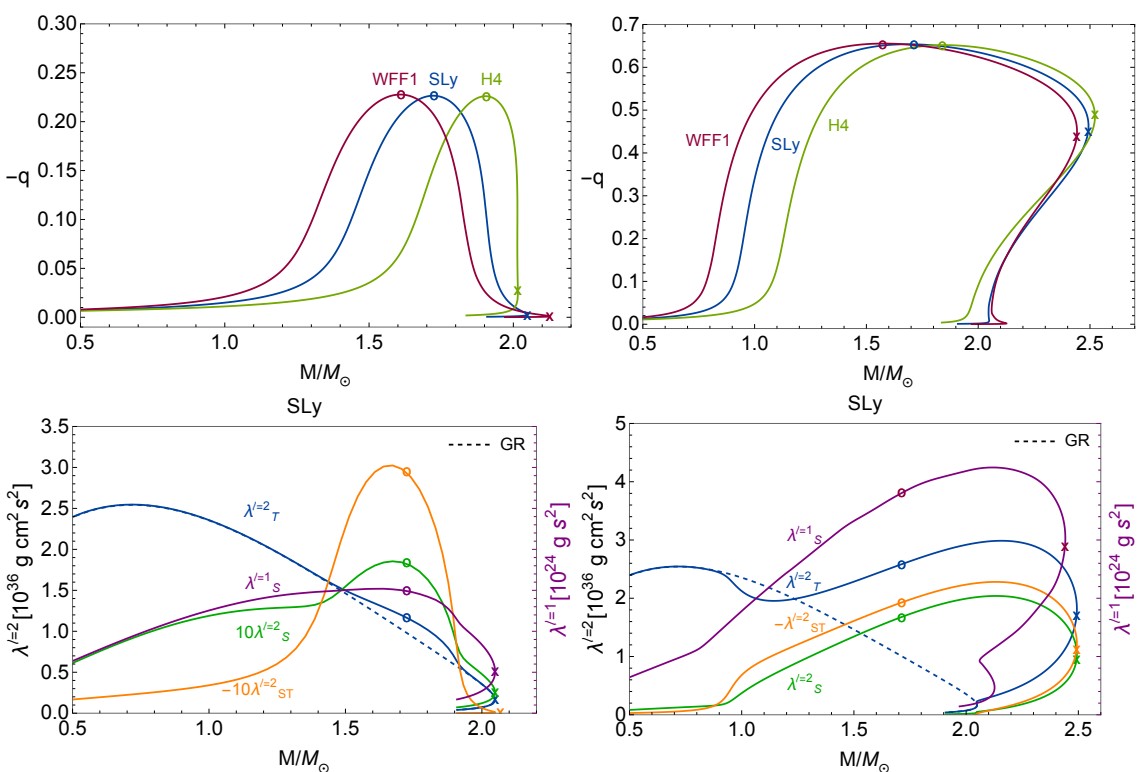

Figure 3: Charge-mass curves (top row), and tidal deformabilties (bottom row) in the Einstein frame for three equations of state (WFF1, SLy, and H4). Quantities are shown for $\beta_0 = -4.5$ (left column) and $\beta_0 = -6$ (right column). Dashed lines represent the GR configurations $\beta_0 = 0$. In the bottom panels we fix the SLy EoS and show the different tidal deformabilities. We note the rescalings of some of the curves in the lower left panel and the different units of the dipolar scalar deformabilities (purple curves and right axes) and the quadrupolar ones (all other curves and left axes). Circles represent the maximum charge configuration, $M_q$, and crosses indicate the maximum mass.

scalarized NS solutions, hence $|\alpha_0|$ is of order $10^{-3}$. This corresponds to the upperbound value on this parameter from solar system tests of the Cassini spacecraft [28, 87, 88]. We do note that this $|\alpha_0|$ and $\beta_0 = -4.5$ are on the lower bound of being consistent with recent binary pulsar tests [24, 27–29]. This does depend on the choice of EoS, where softer EoS allow for a more negative lower bound on $\beta_0$ [28], however $\beta_0 = -6$ can be stated to be ruled out by current observations, regardless of the EoS [29]. Our main goal with these choices for $\beta_0$ is to show the effect and dependencies of this parameter on the tidal contributions to the waveforms for scalarized NSs. For more negative choices of $\beta_0$ the scalarization effects are enhanced and our choices for this parameter are therefore useful for a qualitative study of the parametric dependence. However we highlight that the case studies for $\beta_0 = -4.5$ show the more realistic results.

### 5.1.2 Properties of isolated and tidally perturbed NSs

For the numerical implementation to compute properties of NSs, we use the results from [45], which we briefly summarize.

The properties of NSs are obtained from the field equations derived from (1) with the

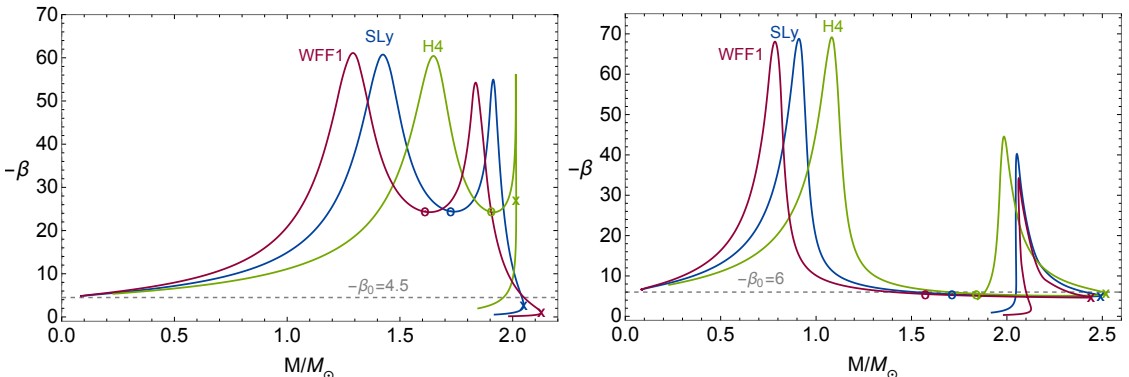

Figure 4: Results for the parameter $\beta$ characterizing the variation of the scalar charge with the field at its cosmological value as a function of mass. The results are computed in the Einstein frame for three EoSs (WFF1, SLy, and H4) and for $\beta_0 = -4.5$ (left column) and $\beta_0 = -6$ (right column) for cosmological scalar field values of $\varphi_\infty = 10^{-3}$. Dashed lines represent the values of the coupling coefficient $\beta_0$, circles indicate the maximum-charge configuration $M_q$, and crosses the maximum mass.

matter action corresponding to a perfect-fluid energy-momentum tensor for a NS, together with stress-energy conservation. To solve the system requires specifying an EoS, for which we use a parameterized piecewise polytropic approximation to tabulated models [89] known as WFF1, SLy and H4. These EoS cover a range of possibilities from softer to stiffer EoSs that, for a given NS mass, lead a corrresponding range of more or less compact stars respectively. To compute these properties we solve the equations of motion numerically using a shooting method, as described in [45]. Specializing the computations to an equilibrium configuration yields the mass, radius, and scalar charge of the NS. The upper panels of Fig. 3 show the results for the scalar charge for $\beta_0 = -4.5$ in the left panel and $\beta_0 = -6$ in the right panel. Each point corresponds to a different central density of the NS that increases from left to right along each curve. The steep rise in charge seen above a certain mass indicates the formation of a significant scalar condensate, i.e. the scalarization of the NS which occurs above a critical compactness, which is reached for lower masses in the case of softer EoSs. We further consider linear, static perturbations to this equilibrium configuration, which enables us to solve for the various tidal deformability parameters shown in the lower panels of Fig. 3 for a fixed intermediate SLy EoS. Here, the $\ell = 2$ scalar (S) and scalar-tensor (ST) values are rescaled by an order of magnitude as they would otherwise be too small to be visible on the plot, and in addition, the sign of the ST results is reversed. We see that in the case of $\beta_0 = -4.5$ (lower left panel), the tensor deformability (blue curves) is identical to its value in GR (dashed lines) for most of the mass range and only differs slightly for masses corresponding to a large scalarization. All other deformability parameters are much smaller than the tensor one throughout the mass range and also exhibit a significant enhancement for large scalarization. For larger values of $\beta_0$ (lower right panel), all effects of scalarization and scalar tides are larger, with all Love numbers attaining the same order of magnitude, however, the quadrupole tensor deformability also dominates over the others in most of the mass range.

## 5.2 Computation of $\beta(\varphi)$

To compute the skeletonized mass (25) requires not only information on the scalar charge, but also how it varies with the cosmological value of the scalar field, as parameterized by $\beta$ in (25). This quantity was not computed in [45]. For objects with negligible self-gravity one can use that [18] $\beta = (d^2 \log A / d\varphi^2)_\infty$, where the evaluation is at the value of the scalar field

609  at infinity. However, compact objects such as NSs have strong self-gravity and this formula does
610  not apply. Thus, we compute $\beta$ numerically by calculating the scalar charge $q$ for a wide range
611  of asymptotic scalar field values, interpolating the results to obtain an approximate functional
612  dependence on the field, and computing the numerical derivative of this interpolating function.
613  Our implementation is based on an extension of the publicly available `Mathematica` code
614  of [45, 90], which we use to generate data around the value of the desired cosmological scalar
615  field $\varphi_\infty = 10^{-3}$. We compute results for $\varphi_\infty = [0.9, 1.1] \times 10^{-3}$ in increments of $0.01 \times 10^{-3}$.
616  We then interpolate the datapoints and differentiate the interpolation function to obtain $\beta$
617  from (25b) with $\varphi_0 = \varphi_\infty$. We also compare this to results obtained directly approximating
618  the numerical derivative by

$$\beta(\varphi_\infty) \approx \frac{q(\varphi_\infty + \Delta\varphi_\infty) - q(\varphi_\infty)}{\Delta\varphi_\infty} \, , \tag{90}$$

619  with $\Delta\varphi_\infty$ an infinitesimal increment. We found that taking $\Delta\varphi_\infty = 0.01 \times 10^{-3}$ is sufficiently
620  small to resolve the derivative, while larger increments of $0.05 \times 10^{-3}$ led to inconsistencies.
621  With this setting, we obtain excellent agreement between the methods, with a maximum frac-
622  tional difference between the approximation to the numerical derivative and the interpolation
623  of 0.37%. This is expected as the dependence of $q$ on the scalar field is smooth. Fig. 4 shows
624  the results for $\beta$ using the approximation to the numerical derivative (90). As a check on
625  our calculations, we also verified that for $M \to 0$ we recover the analytical results for neg-
626  ligible self-gravity. We see that as a function of the NS mass, the maximum $|\beta|$ occurs for
627  relatively low masses and takes a low value for the maximum scalar charge configuration $M_q$.
628  For $\beta_0 = -4.5$ shown in the left panel of Fig. 4, the value of $|\beta|$ for the $M_q$ configuration is a
629  local minimum, and rises to larger values for higher masses before rapidly decreasing and for
630  softer EoSs (WFF1, SLy shown as the red and blue curves respectively) even dropping below
631  $|\beta_0|$ near the maximum mass. By contrast, for the stiffer H4 EoS (green curve), $|\beta|$ peaks
632  abruptly around the maximum mass and remains well above $|\beta_0|$. Such a behavior was also
633  noticed in [91], p. 530, and in Figure 3 of [15]. The behavior of $|\beta|$ is different for $\beta_0 = -6$
634  shown in the right panel of Fig. 4. In that case, all EoSs lead to a maximum value at low
635  masses, fall off below $|\beta_0|$ near the maximum mass, and attain a secondary local maximum for
636  NSs with central densities just above the maximum mass configuration. We note that while for
637  NSs in GR, the maximum mass usually correlates with the onset of a gravitational instability,
638  this is a priori not necessarily the case for the ST configurations considered here. We leave the
639  stability analysis and hence the question if the behavior of $\beta$ beyond the maximum mass is of
640  physical significance to future work.

## 5.3   Waveform and frequency evolution of example binary systems

642  To gain intuition on the impact of tidal effects on the waveforms, we first consider in Fig. 5 the
643  plus polarization of the time-domain waveforms (103) and (57) with the phase evolution com-
644  puted from (67) for a BH-NS binary system with masses $(5, 1.7)M_\odot$ (top panel) and a NS-NS
645  system with masses $(1, 1.7)M_\odot$ (middle panel). In addition, the bottom panel shows the corre-
646  sponding frequency evolution for the NS-NS system. The scalar charge and tidal deformability
647  parameters are taken from the results shown in Fig. 3 for the SLy EoS and $\beta_0 = -4.5$ for NSs
648  and set to zero for BHs. As the PN approximation is not valid for frequencies close to merger
649  we cut off the functions at a benchmark GW frequency of 500Hz[3] and show a snapshot of the
650  waveforms in the top two panels of Fig. 5 around 100Hz. The curves in Fig. 5 correspond
651  to the evolutions of only the point-particle contributions and for point-particle plus tidal con-
652  tributions, both in ST theories. We see that tidal effects are more pronounced for the NS-NS

---

[3]merger frequencies for NS-NS systems are usually around order $10^3$Hz.

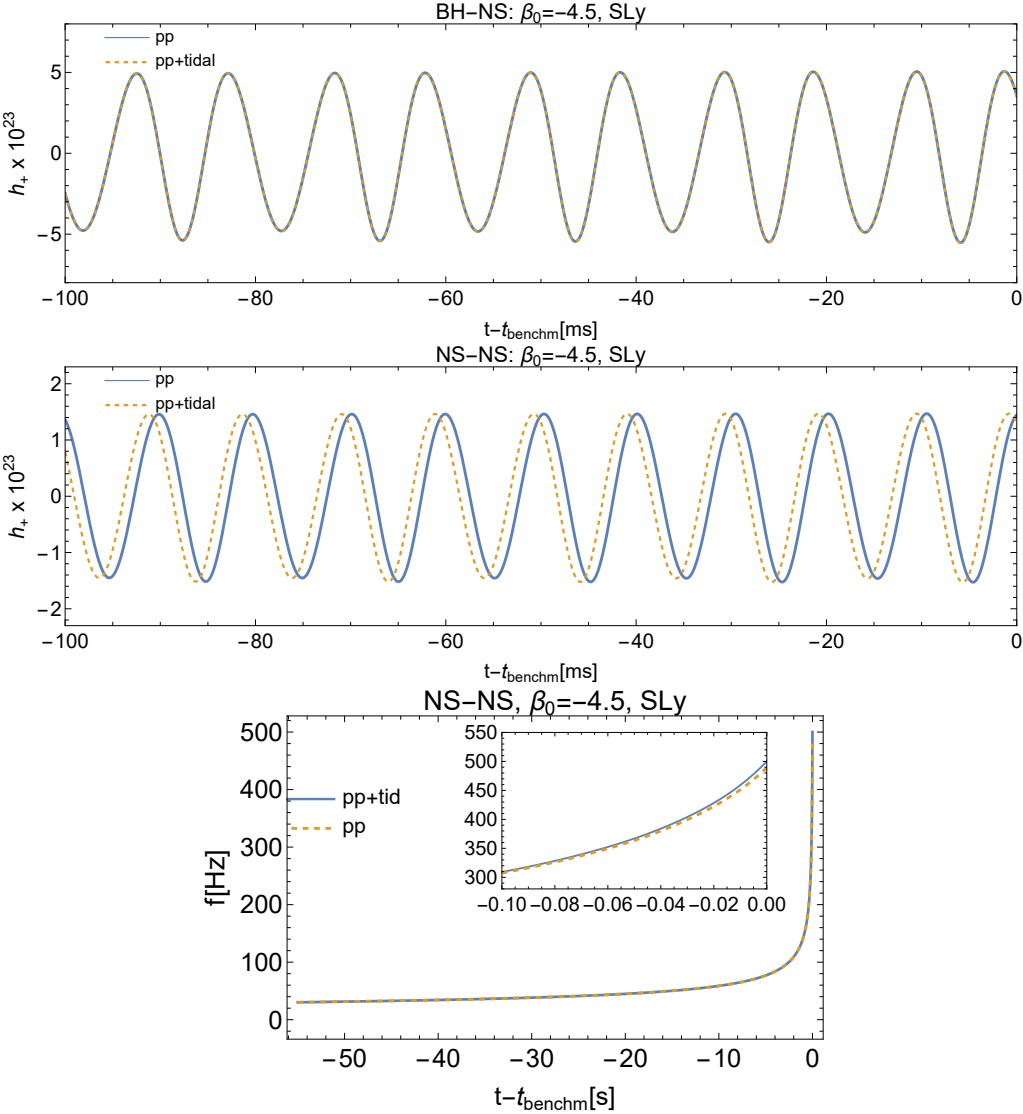

Figure 5: The plus polarization of the time-domain waveform for a BH-NS system (top), NS-NS system (middle) and frequency evolution for the respective NS-NS system (bottom), assuming $\beta_0 = -4.5$ and the SLy EoS. The benchmark time $t_{\mathrm{benchm}}$ corresponds to the a frequency of $f_{\mathrm{benchm}} = 100\mathrm{Hz}$ for the top two panels and $f_{\mathrm{benchm}} = 500\mathrm{Hz}$ for the bottom panel. The inset in the bottom panel shows a zoom in of the curves near the benchmark time.

than for the BH-NS system. The two curves for the NS-NS systems for low frequencies overlap. When the frequency starts to rapidly increase as shown in the bottom panel of Fig. 5, the tidal effects become more noticeable as can be seen in the phase difference between the curves in the middle panel for a NS-NS system. In principle, one might expect the scalar tidal effects to be larger for a BH-NS system, since a large difference between scalar charges maximises the dipolar flux (62). However, when one of the scalar charges vanishes, so do some of the tidal contributions to the phase, as a consequence of $\zeta_1 = \bar{\zeta}_1 = \tilde{\zeta}_1 = 0$ in this case, where the coefficients were defined in (31), (54), and (45). Furthermore we see from Fig. 5 that the effect of the tidal contributions mostly enters in the phase accumulation, while their effect on the amplitude is very small. The more irregular shape of the oscillations for the BH-NS case is due to more power being carried in modes besides the $(2,2)$ mode for more asymmetric masses.

### 5.4 Identifying interesting parameter regimes

To gain insight into the importance of different tidal GW signatures we now focus on the Fourier phase evolution (81). We first survey a large part of the parameter space of NS-NS and BH-NS systems to analyse what systems are particularly interesting with respect to tidal effects on GWs in ST theory. Henceforth in the analysis we mainly focus on the intermediate SLy EoS and $\beta_0 = -4.5$ and comment on the dependency of the results on the EoS and $\beta$ throughout the discussion.

For these parameter space studies we restrict to NS masses between $1 - 2M_\odot$, that is, roughly around the lightest and heavier observed NSs [92], and BH masses between $5-15M_\odot$, with the lower limit representing roughly the lowest mass of 'unambiguous BH candidates' from GW observations discussed in [3].

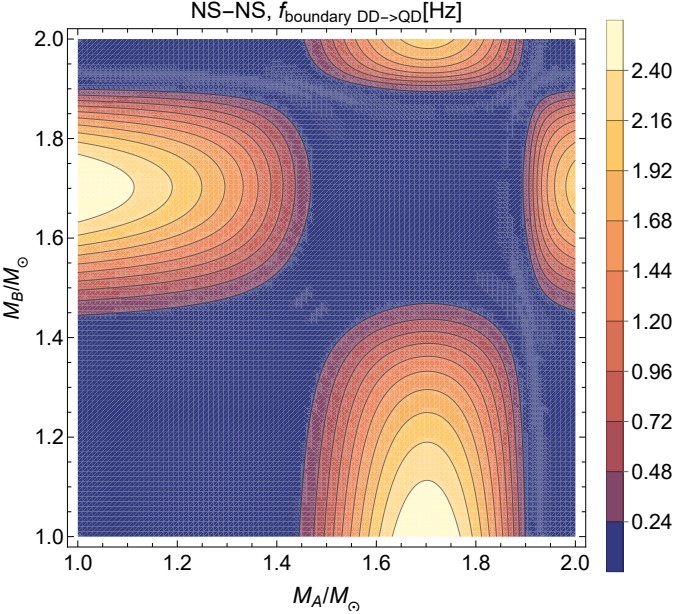

Figure 6: Estimated transition frequency (71) for a NS-NS system between dipole- and quadrupole-driven regimes of the inspiral in the $M_A - M_B$ parameter space for $\beta_0 = -4.5$ and the SLy EoS. The color scale indicates the value of the estimated transition frequency in Hz.

### 5.4.1 Estimated transition between DD and QD domains

We next consider the transition frequency between the regimes dominated by the dipole- and quadrupole losses discussed in Sec. 4.5. Figure 6 shows the estimated transition frequency (71) for a wide parameter space of NS-NS systems. The dependence of (71) on $S_-$ indicates that the transition frequency is highest when the difference between the two scalar charges is largest, as is also confirmed by the numerical sweep of the parameter space shown in Figure 6. The largest charge difference occurs in NS-NS systems when one NS has the maximum charge for which the corresponding mass $M_q$ is identified from Fig. 3 and the companion NS is near the minimum mass within the range considered. Notably, we find that even for NS-NS systems with the largest $S_-$, the frequency where the dipole dominates is $f_{\text{boundary}}^{DD} \lesssim 3\text{Hz}$, while for BH-NS binaries it is even lower due to their larger total mass. This implies that the $DD$ regime lies below the lower end of the sensitive frequency band $f_{\text{low}} \sim 10\text{Hz}$ of current detectors [93] and even that of next-generation ground based GW detectors [94, 95]. Although (71) provides only a rough estimate of an extended range of frequencies where the transition occurs [49], these considerations motivate us to focus on the $QD$ domain in the main part of this paper and delegate results for the $DD$ regime to Appendix F. We also note that the $DD$ regime has potential direct relevance for future deci-Hertz GW detectors, e.g. [96].

### 5.4.2 Tidal Contributions to the QD Fourier phase

Next, we analyze the tidal contributions to the phase by computing the parameter dependencies of the various coefficients in (82). The results are summarized in Table 1. We find that most of the coefficients, specifically $S_-c_2, c_3$, and $S_-^2 c_4$, are strongly correlated with the scalar charge, while by contrast, the $\ell = 2$ non-dipolar coefficient involving the combination of $\tilde{\Lambda}$ and $c_5$ depends most strongly on the total mass. Depending on the system, the latter contribution also has the largest values of all tidal coefficients in the GW phase considered here.

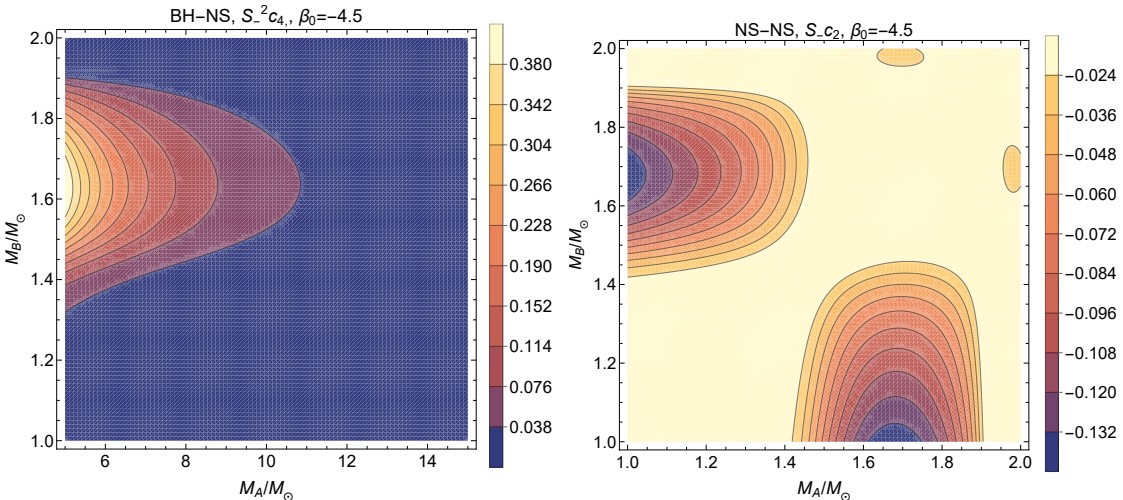

Figure 7: Contour plots of two of the coefficients $S_-c_2$ and $c_3$ defined in (82), in the $M_A - M_B$ parameter space for $\beta_0 = -4.5$ and the SLy EoS for a BH-NS system (top) and NS-NS system (bottom). The plots for the coefficients of other powers of $x$ in the phase can be found in Appendix E.

Figure 7 shows examples of the dipolar contributions to tidal coefficients in (82). The upper panel is the dipolar piece of the $\ell = 2$ tidal terms for BH-NS systems characterized by $c_4 S_-^2$, while the bottom panel shows the dipolar piece of the $\ell = 1$ coefficient, i.e. $c_2 S_-$,

for NS-NS systems. We re-iterate that as discussed above, in BH-NS systems, all $\ell = 1$ tidal effects vanish because they are purely scalar interactions and as the BH has no scalar charge, there is no scalar tidal field felt by the NS. However, we note that even for BH-NS, there is a nonvanishing dipolar contribution to the $\ell = 2$ tidal effects in the phase that arises from the energy fluxes.

For both types of binary systems, a comparison to Fig. 3 shows that the dipolar coefficients $c_2 S_-$ and $c_4 S_-^2$ are largest when (one of) the NS(s) has a mass $M_q$ corresponding to the configuration with the maximum scalar charge and the companion has the lowest mass $M_{\min}$ within the range considered. As the dipolar contributions to (80) are directly proportional to $S_-$, the strong dependence on the scalar charge makes sense, maximizing $S_-$ when the difference between scalar charges is largest. We also see from Fig. 7 that the sign of the dipolar tidal coefficient is negative for $\ell = 1$ (i.e. $S_- c_2$) but positive for $\ell = 2$ (i.e. $S_-^2 c_4$). The analysis of all contributions to (82) in Appendix E further shows that for NS-NS systems, the magnitude of the $\ell = 2$ dipolar coefficient is about three orders of magnitude larger than for the $\ell = 1$ case.

In addition, in Appendix E we find that similarly to the dipolar contributions discussed above, the non-dipolar coefficient $c_3$ is dominated by the dependence on the scalar charge and maximized for $(M_q, M_{\min})$ systems, however, it also has large values for systems around $(M_q, M_q)$. This is because although $c_3$ has no explicit dependence on $S_-$, it still involves $\zeta_1 \propto q_A^2 \lambda_{S,B}^{\ell=1} + (A \leftrightarrow B)$, which becomes large for maximized scalar charge systems around $(M_q, M_q)$. Overall, the values of $c_3$ for the considered parameter space are positive and of order $10^1 - 10^2$.

By contrast, the non-dipolar quadrupolar coefficient characterizing the relative $O(x^5)$ contributions to the phase (82) shows little correlations with the scalar charge but instead depends most strongly on the mass, with the maximum value occurring for the lowest total mass. This tidal coefficient is negative and of order $10^4 - 10^5$ for the parameter space considered here. For BH-NS systems, the qualitative behavior is similar but the overall magnitude is lower $\sim 10^2 - 10^3$ because of the larger total mass of these systems. These results indicate that the tensor tidal contributions (that enter as terms of the form $M_B \lambda_{T,A}^{\ell=2} + (A \leftrightarrow B)$ in the parameter $\tilde{\zeta}_2$ and the phase (80)) to the phase dominate, as also expected from the tidal deformability curves of Fig. 3. This too is reflected in the overall dominant magnitude of the $O(x^5)$ coefficient compared to the other tidal terms in (82), which is also the only nonvanishing contribution in the limit of GR.

Based on the above analysis of the parameter space of tidal coefficients, we choose three representative cases for each type of binary system for further analysis: one each that maximizes the $\ell = 1, 2$ tidal effects respectively, and another where all tidal coefficients take intermediate values (though note that as explained above, some of them are always zero for NS-BH). These choices are listed in Table 2.

Scenario $S1$ maximizes the contribution of the quadrupolar contributions involving $\tilde{\Lambda}$ and $c_5$ and is free from dipolar contributions, since they vanish for an equal NS case as $S_- = 0$. Scenario $S2$ maximizes the effects of the dipolar contributions proportional to $S_-$ together with the non-dipolar scalar tidal effects encapsulated in $c_3$, and Scenario $S3$ is an intermediate case, close but not equal to the identical-NS limit, in order to avoid cancellations of terms scaling as $(q_A - q_B)$.

## 5.5 Tidal effects on the GW phase evolution

To analyze the effects of different tidal contributions on the GWs, we focus on the Fourier domain phase evolution in the QD domain.

| contribution to phase coefficients | largest for | zero for | 2* sign | magnitude for NS-NS ($\log_{10}$) |
|---|---|---|---|---|
| $c_2 S_-$ (scalar $\ell = 1$) | 2*$(M_q, 1)$ | $A = B$, BH-NS | 2*– | 2*[-1, -2] |
| $c_3$ (scalar $\ell = 1$) | $(M_q, 1)$, $(M_q, M_q)$ | 2*BH-NS | 2*+ | 2*[1, 2] |
| $c_4 S_-^2$ (all $\ell = 2$) | 2*$(M_q, M_{\min})$ | 2*$A = B$ | 2*+ | 2*[1, 2] |
| $c_5 + 39\tilde{\Lambda}/(2\alpha^5 \xi^2)$ (all $\ell = 2$) | 2*low $M_{\text{tot}}$ | 2*N/A | 2*+ | 2*[4, 5] |

Table 1: Properties of the various tidal coefficients that appear with different powers of the frequency in the Fourier GW phasing (82) for binaries involving at least one NS. The notation $A = B$ refers to the case of two identical NSs. A dependence on $S_-$ signals a contribution that arises through dipolar effects in the GW phase. The parentheses in the first column indicate the kinds of tidal interactions that contribute.

| scenario | tidal terms | binary | $M_A(M_\odot)$ | $M_B(M_\odot)$ |
|---|---|---|---|---|
| 2*S1 | no dipole & max $\ell = 2$ | NS-NS | 1 | 1 |
| | | BH-NS | 5 | 1 |
| 2*S2 | 2*max dipole | NS-NS | 1 | $M_q$ |
| | | BH-NS | 5 | $M_q$ |
| 2*S3 | 2*intermediate | NS-NS | 1.1 | 1.2 |
| | | BH-NS | 8.9 | 1.2 |

Table 2: Properties of binary systems considered for the case studies. We consider three scenarios to cover two extremes and an intermediate case within the parameter space described in the text and with $M_q \sim 1.7 M_\odot$ here.

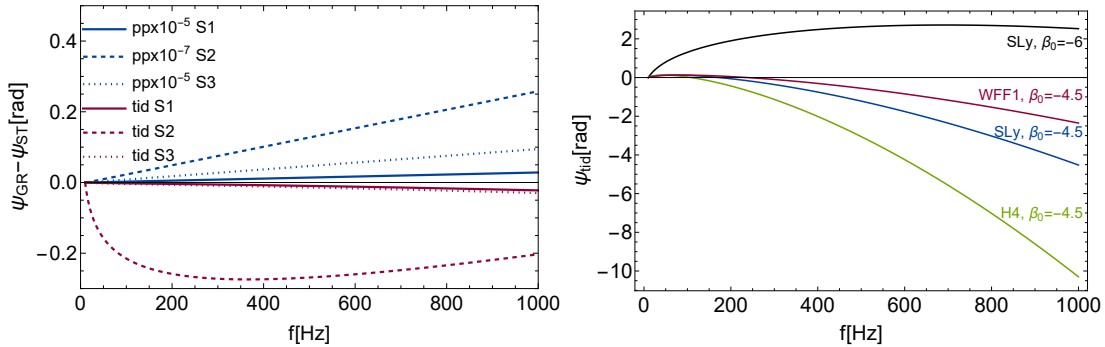

Figure 8: *Left panel*: Difference between GR and ST Fourier phase as a function of frequency for NS-NS binaries. The blue lines correspond to the point particle (pp) contributions and the red ones to the tidal contributions. The solid, dashed and dotted lines indicate the different scenarios S1, S2 and S3 respectively. Note that the point-particle curves are rescaled by many orders of magnitude to show them on the same plot. The pp phase evolutions were aligned in the window of $9 - 10$ Hz. *Right panel*: the tidal contribution of the ST Fourier phase for NS-NS binary of S2 for different EoS and $\beta_0$. The tidal phase functions in both panels were set to zero at 10Hz with the freedom in initial phase angle (80).

### 5.5.1 Difference in net tidal effects between ST and GR

The left panel of Figure 8 shows the difference between the GR Fourier phase and the ST phase in the quadrupolar driven domain (80) including corrections up to relative 1PN order. The evolutions of the point particle phase in ST and GR, denoted by the blue curves, are matched using the freedom in initial conditions for (80) to find the initial orbital phase and time that minimize the integral of the absolute value of the difference squared over the frequency domain $9 - 10$Hz. For the tidal contributions (red curves) we only used the freedom in initial phase angle to set the phase contributions equal to zero at the starting frequency of 10Hz. We chose this approach for the tidal contributions as the dependency on the frequency for the ST tidal contributions is qualitatively different from the GR tidal terms, hence their evolutions as function of the frequency are hard to align over a frequency range. Do note that in the case of the tidal phase difference shown in the left panel of Fig. 8 there is an additional free initial condition $t_0$ which we now set to zero. We find that, as expected, the difference in point particle contributions is many orders of magnitude larger than the tidal contributions, with the largest point-particle differences for Scenario S2, where the dipolar scalar radiation is maximized.

For the tidal contributions, the phase difference between GR and ST is negative, indicating that the net tidal effects in ST theory are smaller than in GR. This is because in GR, only the quadrupolar tensor tidal effects appear and lead to a negative contribution to the phase, indicating that the inspiral is faster and hence accumulates less phase per frequency interval than the point-particle scenario. By contrast, in ST theories, tidal effects in the GW phase include dipolar and quadrupolar scalar, tensor and scalar-tensor contributions having different signs. These combine to yield a total tidal phase correction that is smaller in magnitude than that in GR.

In particular, the main sources of the differences between tidal effects in GR and ST theories are the scalar and scalar-tensor tidal contributions. Scenario S2 maximizes these contributions and we see from the dashed red curve in Fig 8 that it indeed leads to the largest differences. The different shape of the S2 curve in the left panel can be explained together with the blue curve in the right panel. The right panel of Fig. 8 shows the ST tidal fourier phase contributions for different EoS and choices of $\beta_0$ in S2. The blue curve corresponds to the choices of parameters adopted in the left panel. We find that in this scenario, not only do the additional scalar and scalar tensor tidal contributions make the total tidal contributions smaller than in GR, also the sign becomes positive in the frequency window from $0 - 200$Hz. This positive evolution sharply increases the difference with the negative GR tidal phase evolution shown with the dashed curve in the left panel. Beyond 200Hz the ST tidal phase contribution also becomes negative again and starts to decrease its difference with GR. In the other scenarios the ST tidal contributions with a positive sign are less prominent and the total ST tidal phase stays negative over the whole frequency domain, showing a small offset with the GR tidal phase contribution in the left panel. In Scenario S1 (solid red curve) the quadrupolar tensor tidal contributions are maximized, however this contribution differs only slightly from the GR contribution via the small change in the quadrupolar tensor tidal deformability. In this Scenario the other tidal contributions are minimal and therefore also the difference with GR is smallest for this system.

In the right panel of Fig. 8 we show the effect of a stiffer (H4) respectively softer EoS (WFF1) and a larger value for $-\beta_0$ on the ST tidal contributions to the Fourier phase in S2. We find that for a stiffer EoS, the magnitude of the tidal contributions increases, which corresponds to the tidal deformabilities being larger for a stiffer EoS [45]. For a larger value of $-\beta_0$ the positive sign tidal contributions to the total phase are severely amplified leading to a positive tidal phase contribution over the whole frequency range. It is interesting to note that while one might in general expect ST effects for larger $|\beta_0|$ to be larger, we see here that the net tidal effects are of the same order of magnitude, except for the sign difference. This is because

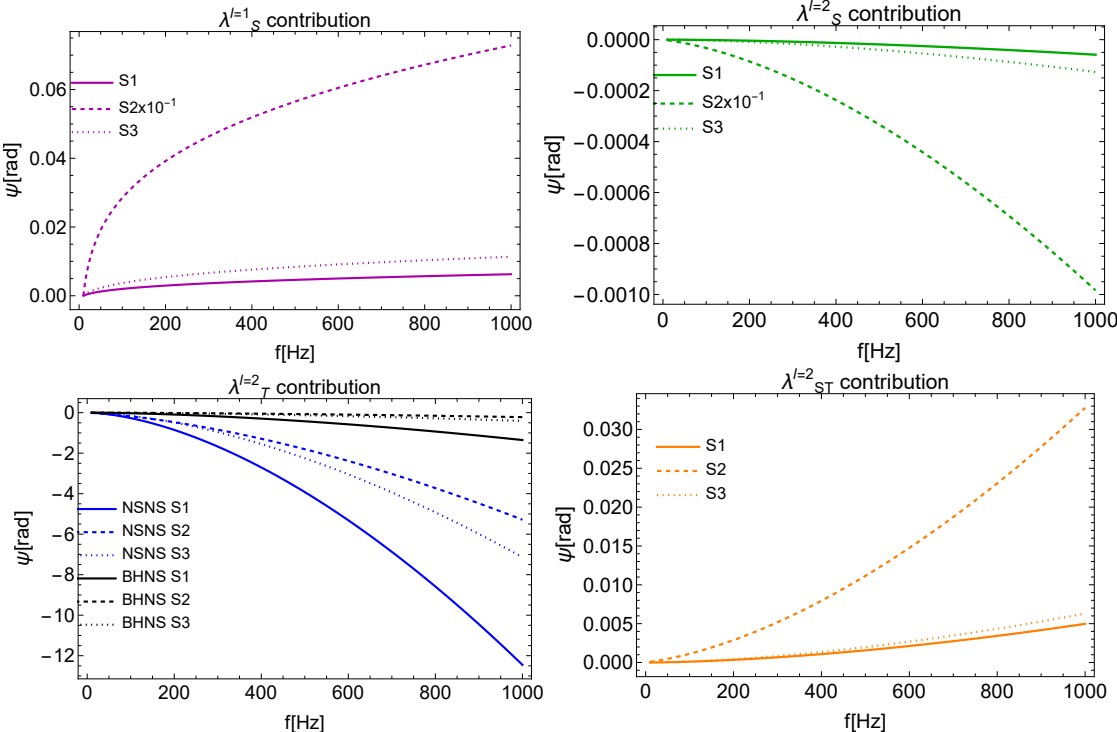

Figure 9: Different contributions of the tidal deformabilities to the Fourier phase
(81). In each panel the solid, dashed and dotted curve correspond to the systems of
Scenario S1, S2 and S3 respectively for the NS-NS systems. The bottom left plot also
shows these scenarios for BH-NS systems in black.

the comparison here is done at fixed NS masses, which correspond to the configurations that maximize the dipole effects for $|\beta_0| = 4.5$ while the maximum dipole configuration for $|\beta_0| = 6$ occurs for different masses.

### 5.5.2 Effect of scalar, tensor, and mixed tidal deformabilities

It is further interesting to consider the effect of the different scalar, tensor, and mixed scalar-tensor tidal interactions on the net tidal phase. Fig. 9 shows the different contributions of the corresponding tidal deformabilities that make up the $c$ parameters of (82) for different scenarios of NS-NS and BH-NS systems for the quadrupolar driven phase tidal effects. To generate these plots, we isolate the terms proportional to the different tidal deformabilities in (80) and (81). We terminate the phase evolution at a chosen benchmark frequency of 1kHz for illustration, although the approximations made to derive our results become invalid before then.

Comparing the panels of Fig. 9 we find that in all studied cases the dipolar ($\ell = 1$) scalar tidal contributions and the quadrupolar ($\ell = 2$) scalar-tensor contributions come with a positive sign and the quadrupolar scalar and quadrupolar tensor with negative sign, meaning that they slow down and speed up the inspiral respectively.

Furthermore, we find for all NS-NS scenarios that the quadrupolar tensor tidal contributions dominate over the others, as was already predicted from the analysis in Sec. 5.4.2. Also, comparing the quadrupolar tensor contributions for the NS-NS and BH-NS cases, we find that the phase contribution in the BH-NS case is around an order of magnitude smaller because of the larger total mass of these systems. Generally, the qualitative behaviour of the tidal contributions is the same for the other EoSs, only the magnitude of the accumulated Fourier phase is slightly shifted, with slightly larger effects for H4 and slightly smaller effects for WFF1 related to the shift in order of magnitude of the tidal deformability which is bigger for the stiffer EoM H4 and smaller for the softer EoS WFF1. For $\beta_0 = -6$ the same conclusion holds however the effects are more prominent as the accumulated phase increases around one order of magnitude. We also see from Fig. 9 that the choices of scenarios summarized in Table 2 have the expected effects, with the largest the tensor quadrupole effects for S1 and the largest dipolar scalar and quadrupolar scalar and scalar-tensor contributions for S2, while S3 described an intermediate case. Hence, we conclude that for studying the tidal effects specifically present in scalar tensor theory (scalar dipolar and quadrupolar scalar and scalar-tensor tidal effects) a system that maximizes the difference between the two scalar charges of the two bodies is preferred. This also corresponds to what we found for the added contribution in Fig. 8.

### 5.5.3 Discussion

The fact that the tidal phase difference with GR in Fig. 8 is negative means that there could be degeneracies between the ST accumulated tidal effects to the phase and the GR tidal contributions for the same system but with a softer EoS also giving smaller tidal contributions. However, the large difference for the point particle contributions would still resolve two situations. Additionally, as the difference in tidal contributions stays below a magnitude of 1 rad they are unlikely to be resolvable with the current GWs detectors. Still the degeneracy with GR might be a problem for other beyond-GR theories with scalarized NSs for which omitting the scalar and scalar-tensor tidal effects could potentially lead to biases in the inferred properties of the compact objects from GW observations, especially for systems with large differences between the scalar charges of the bodies. Furthermore, from comparing the top left and bottom right panels of Fig. 9 the magnitude of the dipolar scalar tidal contributions for S1 and S3 are of the same order as the quadrupolar scalar-tensor contributions, hence when including the scalar tidal contributions the interaction with the tensor field cannot be neglected.

## 6   Summary and Conclusion

In this paper, we studied tidal signatures in GWs from NS binary systems in scalar-tensor theories of gravity where sufficiently compact NSs can give rise to a scalar condensate. Building on [45], which showed that in ST theories, tidal effects are characterized by three different kinds of tidal deformability coefficients for the scalar, tensor, and mixed sector arising from the nonlinear coupling between gravity, scalar field, and baryonic matter, we used analytical approximations in finite-size and PN corrections in the early inspiral to compute tidal effects in the GWs.

We showed that in addition to a tidal term similar to GR, there are also terms that scale with lower powers of the frequency and involve different combinations of tidal coefficients. Specifically, we showed that the Fourier GW phase in the regime of greatest interest for current and next-generation ground-based GW detectors can be written as

$$\psi_{\text{tid}} \quad = \quad \frac{3}{128\eta x^{5/2}}\left[c_2 S_- x^2 + c_3 x^3 + c_4 S_-^2\left(\log x - \frac{2}{3}\right)x^4 + \left(\frac{39}{2\alpha^5\xi^2}\tilde{\Lambda} + c_5\right)x^5\right],$$

where the various coefficients $c_i$ for $i = 2, 3, 4, 5$ are given explicitly in terms of masses, scalar charges, and tidal deformabilities of the bodies in (120)– (123), and $\alpha$ and $\xi$ were defined in (30) and (77), with $S_\pm$ given in (61) and $\tilde{\Lambda}$ having the same functional form (83) in terms of mass-weighted combinations of quadrupolar tensor tidal deformabilities as in GR. We also recall that in ST theories, the frequency-parameter $x$ defined in (36) differs from its GR counterpart by a factor of $\alpha^{2/3}$ corresponding to a renormalization of the gravitational interaction.

We then specialized to a Gaussian coupling function for the ST theory and surveyed binary systems over a range of parameter space to identify interesting scenarios that maximize the different tidal contributions. In general we found that due to different signs associated with the different types of tidal effects in the GW phase (c.f. Table 1), the net tidal signatures in ST gravity are smaller than in GR. This could in principle lead to degeneracies with changes in the EoS of the NSs to a softer one but conclusions about this issue will require further work. We also showed that the difference with GR tidal effects is largest for systems with the largest asymmetry in scalar charges and analyzed the effects of changing the EoS and the theory-parameter $\beta_0$ for a binary system of fixed masses.

By systematically studying each tidal contribution to the Fourier phase, we further demonstrated that, at quadrupolar order, the scalar-tensor tidal deformability has a larger effect on GWs than the scalar tidal deformability and therefore should be accounted for when also considering the scalar tidal deformability. Furthermore, depending on the parameters, the different contributions that come with different powers of frequency can lead to a non-monotonic behavior of the tidal phase evolution as a function of frequency. Quantitatively, the difference in tidal effects on the phase between ST and GR stays of order $10^{-1}$ rad and is therefore difficult to resolve with current detectors. It is also many orders of magnitude smaller than the differences in point-particle inspirals, which would directly indicate modifications to GR and hence reduce potential degeneracies with EoS effects in the small tidal corrections.

While our case studies led to quantitative insights into the impact of the dominant adiabatic tidal phenomena in particular ST theories, our main aim was to develop the methodology for understanding and modeling tidal GW signatures from scalarized compact objects in theories of gravity beyond GR, which lead to richer features. Our methods have broad applications to a wider range of proposed classes of theories, though the details of the GW phase contributions will be theory-dependent. This work opens several avenues for further studies. For instance, our results could be used for data analysis to put constraints on the parameter space covered by ST theories. It would also be interesting to compare our results, combined with higher PN order point-particle terms from [68] to the numerical calculations of BH-NS binaries in ST

theories [51] or others for NS-NS systems, to include a mass of the scalar field as the lowest order self-interaction, to consider dynamical tides, and tidal signatures in GWs in other classes of beyond-GR theories.

# 7 Acknowledgements

We thank Justin Janquart for insightful discussions. G.C., and T.H. acknowledge funding from the Nederlandse Organisatie voor Wetenschappelijk Onderzoek (NWO) sectorplan. I.G. acknowledges the Dutch Black Hole Consortium (project NWA 1292.19.202) part of the National Research Agenda

## A Useful identities involving symmetric trace-free tensors

Here, we summarize several identities used in the derivations discussed in the main text. In this appendix we set $r = |\mathbf{x}|$ with $\mathbf{x}$ a separation vector. The result of taking $\ell$ derivatives of $1/r$ can be written as

$$\partial_L \frac{1}{r} = (-1)^\ell (2\ell - 1)!! \frac{n_{<L>}}{r^{\ell+1}}, \tag{91}$$

where $n = \mathbf{x}/r$ is a unit vector. This was used to obtain explicit expressions for the tidal fields in (27). In the tidal Lagragian (17) we needed the contraction of two tidal tensors and hence STF unit vectors, which simplifies via the identity

$$n_{\langle L \rangle} n^{\langle L \rangle} = \frac{\ell!}{(2\ell - 1)!!}. \tag{92}$$

To derive the tidal contributions to the acceleration required the additional identities

$$\partial_i \left( \partial_L \frac{1}{r} \right)^2 = -2 \frac{(2\ell - 1)!!(\ell + 1)!}{r^{2\ell+3}} n^i, \tag{93}$$

and

$$n_{<L>} n^{<iL>} = \frac{(\ell + 1)!}{(2\ell + 1)!!} n^i. \tag{94}$$

## B Full expressions for the binary dynamics, waveforms and GW phase

In this appendix, we provide the complete expressions for various quantities to 1PN order in the point-mass sector and leading order in tidal effects.

### B.0.1 Binary dynamics

In Sec. 3.1 we discussed how to obtain the leading-order tidal corrections to the binary dynamics. Together with the point particle 1PN contributions, the two-body Lagrangian is given by

$$\begin{aligned}
L_{AB} = &-M_A + \frac{1}{2} M_A \mathbf{v}_A^2 + \frac{G \alpha M_A M_B}{2r} + \left( \frac{1}{8} M_A \mathbf{v}_A^4 \right. \\
&+ \frac{G \alpha M_A M_B}{r} \left[ -\frac{G \alpha M_A}{2r} \left( 1 + 2\bar{\beta}_B \right) + \frac{3}{2} \left( \mathbf{v}_A^2 \right) \right. \\
&\left. -\frac{7}{4} \left( \mathbf{v}_A \cdot \mathbf{v}_B \right) - \frac{1}{4} \left( \mathbf{n} \cdot \mathbf{v}_A \right) \left( \mathbf{n} \cdot \mathbf{v}_B \right) + \frac{\bar{\gamma}}{2} \left( \mathbf{v}_A - \mathbf{v}_B \right)^2 \right] \right) \\
&+ \frac{1}{2} G^2 \mu M \alpha^2 \sum_\ell \frac{(2\ell - 1)!!}{2r^{2(\ell+1)}} \zeta_l + (A \leftrightarrow B),
\end{aligned} \tag{95}$$

where we use boldface to indicate spatial vectors and define

$$\bar{\gamma} \equiv -2 \frac{q_A q_B}{\alpha}, \quad \bar{\beta}_{A/B} \equiv \frac{1}{2} \frac{\beta_{A/B} \, q_{B/A}^2}{\alpha^2}, \tag{96}$$

$$\beta_\pm \equiv \frac{\bar{\beta}_A \pm \bar{\beta}_B}{2}. \tag{97}$$

918 The relative acceleration obtained from (95) is

$$
\begin{aligned}
\mathbf{a} = &-\frac{G\alpha M}{r^2}\mathbf{n} + \frac{G\alpha M}{r^2}\left\{\mathbf{n}\left[\frac{3}{2}\eta\dot{r}^2 - (1+3\eta+\bar{\gamma})\mathbf{v}^2\right]\right.\\
&+ 2\mathbf{v}\dot{r}[2-\eta+\bar{\gamma}] + \frac{2G\alpha M\mathbf{n}}{r}\left[2+\eta+\bar{\gamma}+\beta_+ - \frac{\Delta M}{M}\beta_-\right]\\
&\left.- G^2\alpha^2 M\sum_\ell \frac{(2\ell-1)!!(\ell+1)}{r^{2\ell+3}}\mathbf{n}\zeta_\ell\right\}.
\end{aligned}
\tag{98}
$$

919 The radial component of the equations of motion (98) for circular orbits with $\dot{r} = \ddot{r} = 0$
920 lead to the angular frequency

$$
\begin{aligned}
\omega^2 = \frac{G\alpha M}{r^3}&\left[1 - \frac{\alpha M}{r}\left(3-\eta-\frac{2\beta_-\Delta M}{M}+2\beta_+\right.\right.\\
&\left.\left.+\bar{\gamma}-\sum_\ell \frac{(\ell+1)(2\ell-1)!!}{r^{2\ell}M}\zeta_\ell\right)\right],
\end{aligned}
\tag{99}
$$

921 with $\Delta M \equiv M_A - M_B$. Inverting this expression perturbatively and expressing $\omega$ in terms of
922 the PN parameter $x$ defined in (36) yields

$$
\begin{aligned}
r(x) \quad = \quad \frac{G\alpha M}{x}&\left[1 - \frac{1}{3}x\left(3-\eta+\bar{\gamma}+2\beta_+ - 2\frac{\Delta M}{M}\beta_-\right.\right.\\
&\left.\left.+\sum_\ell \frac{(2\ell-1)!!(\ell+1)}{G^{2\ell}\alpha^{2\ell}M^{1+2\ell}}x^{2\ell}\zeta_\ell\right)\right].
\end{aligned}
\tag{100}
$$

923 Using this result and the energetics from the Lagrangian (95) we obtain for the binding energy
924 to the orders of approximation we are considering

$$
\begin{aligned}
E(x) = -\frac{\mu x}{2}&\left[1 + \frac{2}{3}x\left(\beta_+ - \frac{\Delta M}{M}\beta_- - \bar{\gamma} - \frac{9+\eta}{8}\right)\right.\\
&\left.- \frac{1}{3}\sum_\ell (2\ell-1)!!(4\ell+1)\frac{G^{-2\ell}\zeta_\ell}{M^{2\ell+1}\alpha^{2\ell}}x^{2\ell+1}\right].
\end{aligned}
$$

### B.0.2 Scalar and tensor waves

926 In Sec. 4 we discuss how one obtains the Newtonian tidal corrections to the scalar and tensor
927 waveform. Together with the 1PN point particle terms the scalar waveform is given by

$$
\delta\varphi = \frac{G\mu\sqrt{\alpha}}{d}\left\{P^{-1/2}\tilde{\Phi} + \tilde{\Phi} + P^{1/2}\tilde{\Phi} + P^{1/2}\tilde{\Phi}_{tid}\right\},
\tag{101}
$$

here the superscript of $P$ denotes the PN order of the coefficients. The coefficients are given by

$$
\begin{aligned}
P^{-1/2}\tilde{\Phi} =& 2\mathcal{S}_-(\mathbf{n}\cdot\mathbf{v}) \\
\tilde{\Phi} =& \left(\mathcal{S}_+ - \frac{\Delta M}{M}\mathcal{S}_-\right)\left[-\frac{G\alpha M}{r}\left(\frac{\mathbf{n}\cdot\mathbf{r}}{r}\right)^2 + (\mathbf{n}\cdot\mathbf{v})^2 - \frac{1}{2}v^2\right] + \frac{G\alpha M}{r}\left[-2\mathcal{S}_+ + \frac{8}{\bar{\gamma}}(\mathcal{S}_+\beta_+ + \mathcal{S}_-\beta_-)\right], \\
P^{1/2}\tilde{\Phi} =& \left(-\frac{\Delta M}{M}\mathcal{S}_+ + (1-2\eta)\mathcal{S}_-\right)\left[\frac{3}{2}\frac{G\alpha M}{r^4}\dot{r}(\mathbf{n}\cdot\mathbf{r})^3 - \frac{7}{2}\frac{G\alpha M}{r^3}(\mathbf{n}\cdot\mathbf{v})(\mathbf{n}\cdot\mathbf{r})^2 + (\mathbf{n}\cdot\mathbf{v})^3\right] \\
& + (\mathbf{n}\cdot\mathbf{v})\left\{\left(\frac{\Delta M}{M}\mathcal{S}_+ - \eta\mathcal{S}_-\right)v^2 + \frac{G\alpha M}{r}\left[\frac{1}{2}\frac{\Delta M}{M}\mathcal{S}_+ + \left(2\eta - \frac{3}{2}\right)\mathcal{S}_-\right.\right. \\
& \left.\left. - \frac{4}{\bar{\gamma}}\frac{\Delta M}{M}(\mathcal{S}_+\beta_+ + \mathcal{S}_-\beta_-) + \frac{4}{\bar{\gamma}}(\mathcal{S}_-\beta_+ + \mathcal{S}_+\beta_-)\right]\right\} \\
& + \frac{G\alpha M}{r^2}\dot{r}(\mathbf{n}\cdot\mathbf{r})\left[\frac{3}{2}\mathcal{S}_- - \frac{5}{2}\frac{\Delta M}{M}\mathcal{S}_+ + \frac{4}{\bar{\gamma}}\frac{\Delta M}{M}(\mathcal{S}_+\beta_+ + \mathcal{S}_-\beta_-) - \frac{4}{\bar{\gamma}}(\mathcal{S}_-\beta_+ + \mathcal{S}_+\beta_-)\right] \\
P^{1/2}\tilde{\Phi}_{tid} =& \sum_\ell\sum_{k=0}^{\ell}\sum_{p=1}^{k}\frac{GN^L\bar{\zeta}_\ell}{\sqrt{\alpha}\mu}\frac{(2\ell+p)!(-1)^{p+\ell}}{k!(\ell-k)!2^\ell}\times\frac{\partial_t^{\ell-k}r_{<L>}}{r^{2\ell+p+1}}B_{k,p}(\dot{r},\ddot{r},\ldots,r^{k-p+1}).
\end{aligned}
$$
(102)

The tensor waveform has the form

$$
h_{TT}^{ij} = \frac{2G\mu}{d}\{Q^{ij}\}_{TT} = \frac{2G\mu}{d}\left\{\tilde{Q}^{ij} + P^{1/2}\tilde{Q}^{ij} + \left(P\tilde{Q}^{ij} + P\tilde{Q}_{tid}^{ij}\right)\right\}_{TT}
$$
(103)

with $\{\ldots\}_{TT}$ denoting the TT projection and

$$
\begin{aligned}
\tilde{Q}^{ij} =& 2\left[v^{ij} - \frac{GM\alpha r^{ij}}{r^3}\right], \\
P^{1/2}\tilde{Q}^{ij} =& \frac{\Delta M}{M}\left[3\frac{GM\alpha}{r^3}(\hat{\mathbf{n}}\cdot\mathbf{r})\left(2v^{(i}r^{j)} - \frac{\dot{r}r^{ij}}{r}\right) - (\hat{\mathbf{n}}\cdot\mathbf{v})\left(2v^{ij} - \frac{GM\alpha r^{ij}}{r^3}\right)\right], \\
P\tilde{Q}^{ij} =& \frac{1-3\eta}{3}\left\{(\mathbf{r}\cdot\hat{\mathbf{n}})^2\frac{GM\alpha}{r^3}\left[\left(6\bar{E} - 15\dot{r}^2 + 13\frac{GM\alpha}{r}\right)\frac{r^{ij}}{r^2} + 30\dot{r}\frac{r^{(i}v^{j)}}{r} - 14v^{ij}\right]\right. \\
& \left. + (\hat{\mathbf{n}}\cdot\mathbf{v})^2\left[6v^{ij} - 2\frac{GM\alpha}{r^3}r^{ij}\right] + \frac{1}{2}(\mathbf{r}\cdot\hat{\mathbf{n}})(\hat{\mathbf{n}}\cdot\mathbf{v})\frac{GM\alpha}{r^2}\left[12\frac{\dot{r}r^{ij}}{r^2} - 32\frac{r^{(i}v^{j)}}{r}\right]\right\} \\
& + \frac{1}{3}\left\{\left[3(1-3\eta)v^2 - 2(2-3\eta)\frac{GM\alpha}{r}\right]v^{ij} + 4\frac{GM\alpha}{r}(5+3\eta+3\bar{\gamma})\frac{\dot{r}}{r}r^{(i,v^{j)}}\right. \\
& \left. + \frac{GM\alpha}{r^3}r^{ij}\left[3(1-3\eta)\dot{r}^2 - (10+3\eta+6\bar{\gamma})v^2 + \left(29+12\bar{\gamma}+12\beta_+ - 12\frac{\Delta M}{M}\beta_-\right)\frac{GM\alpha}{r}\right]\right\}, \\
P\tilde{Q}_{\text{tid}}^{ij} =& -\sum_\ell\frac{2G^2\alpha^2M^2(1+\ell)(2\ell-1)!!\zeta_\ell}{r^{2(2+\ell)}}r^ir^j \\
& + \sum_{\ell=2}^{\ell}\sum_{k=0}^{\ell}\sum_{p=1}^{k}\frac{2GN^{L-2}\tilde{\zeta}_\ell}{\mu}\frac{(2\ell+p)!(-1)^{p+\ell}}{k!(\ell-k)!2^\ell}\frac{\partial_t^{\ell-k}r_{<L>}}{r^{2\ell+p+1}}B_{k,p}(\dot{r},\ddot{r},\ldots,r^{k-p+1}),
\end{aligned}
$$

with $\bar{E} = v^2/2 - Gm\alpha/r$ the leading order binding energy in CM coordinates [65, 75]. In Sec. 4.4 we take the angular integral of the square of the time derivative to the waveforms to

934    obtain the scalar and tensor energy flux. The scalar flux up to 1PN is given by

$$
\begin{aligned}
\mathcal{F}_S = \frac{\eta^2}{G\alpha}\left(\frac{G\alpha M}{r}\right)^4 & \bigg[\frac{4}{3}\mathcal{S}_-^2 + \frac{8}{15}\bigg(\frac{G\alpha M}{r}\bigg[\bigg(-23 + \eta - 10\bar{\gamma} - 10\beta_+ + 10\frac{\Delta M}{M}\beta_-\bigg)\mathcal{S}_-^2 \\
& -2\frac{\Delta M}{M}\mathcal{S}_+\mathcal{S}_-\bigg] + v^2\bigg[+2\mathcal{S}_+^2 + 2\frac{\Delta M}{M}\mathcal{S}_+\mathcal{S}_- + (6 - \eta + 5\bar{\gamma})\mathcal{S}_-^2 - \frac{10}{\bar{\gamma}}\frac{\Delta M}{M}\mathcal{S}_-(\mathcal{S}_+\beta_+ + \mathcal{S}_-\beta_-) \\
& +\frac{10}{\bar{\gamma}}\mathcal{S}_-(\mathcal{S}_-\beta_+ + \mathcal{S}_+\beta_-)\bigg] + \dot{r}^2\bigg[+\frac{23}{2}\mathcal{S}_+^2 - 8\frac{\Delta M}{M}\mathcal{S}_+\mathcal{S}_- + \bigg(9\eta - \frac{37}{2} - 10\bar{\gamma}\bigg)\mathcal{S}_-^2 - \frac{80}{\bar{\gamma}}\mathcal{S}_+(\mathcal{S}_+\beta_+ \\
& +\mathcal{S}_-\beta_-) + \frac{30}{\bar{\gamma}}\frac{\Delta M}{M}\mathcal{S}_-(\mathcal{S}_+\beta_+ + \mathcal{S}_-\beta_-) - \frac{10}{\bar{\gamma}}\mathcal{S}_-(\mathcal{S}_-\beta_+ + \mathcal{S}_+\beta_-) + \frac{120}{\bar{\gamma}^2}(\mathcal{S}_+\beta_+ + \mathcal{S}_-\beta_-)^2\bigg]\bigg) \\
& +\frac{4S_-}{3M^2\eta\alpha^{3/2}r^2}\bigg(9\dot{r} - 3v^2 + \frac{2GM\alpha}{r}\bigg)\bar{\zeta}_1 + \frac{8GS_-^2\alpha}{3r^3}\bigg(2\zeta_1 + 9\frac{\zeta_2}{r^2}\bigg)\bigg].
\end{aligned}
\tag{104}
$$

935    Assuming circular orbits we can write the scalar flux in terms of PN parameter $x$ defined
936    in (36) as

$$
\mathcal{F}_S(x) = x^4\big[S4 + x\big(S5 + S5_{\text{tid},1}x^2 + S5_{\text{tid},2}x^4\big)\big],
\tag{105}
$$

937    with

$$
\begin{aligned}
S4 =& \frac{4\eta^2 S_-^2}{3\alpha G}, \\
S5 =& \bigg(\frac{8\eta^2 S_-}{45\alpha G}\bigg)\bigg(\frac{15}{4\alpha^{3/2}M}(\Delta M q_B\beta_A - M q_B\beta_A + \Delta M q_A\beta_B + M q_A\beta_B) + 10\frac{\Delta M}{M}S_-\beta_- \\
& - S_-(5\bar{\gamma} + 10\beta_+ + 10\eta + 21) + \frac{6S_+^2}{S_-}\bigg), \\
S5_{\text{tid},1} =& \bigg(-\frac{4S_-\mu\bar{\zeta}_1}{3\alpha^{9/2}G^3 M^5} + \frac{16S_-^2\mu^2\zeta_1}{9\alpha^3 G^3 M^5}\bigg), \\
S5_{\text{tid},2} =& \frac{S_-^2\mu^2\zeta_2}{\alpha^5 G^5 M^7}.
\end{aligned}
\tag{106}
$$

938    Note that our notation here differs from [64, 65, 67] as we have the expressed several terms
939    in S5 explicitly in terms of the scalar charge $q$, while [64, 65, 67] rewrite them as a term
940    proportional to $1/\bar{\gamma}$. This has the drawback that for a BH-NS system, with the black hole
941    having zero scalar charge, the parameter $\bar{\gamma}$ vanishes and causes an apparent divergence in the
942    expressions from [64, 65, 67]. Here, we have explicitly expanded the dependencies on $1/\bar{\gamma}$ in
943    terms of the constituent parameters such as the scalar charge and $\beta$, which leads to manifestly
944    finite expressions in the limit of vanishing scalar charge.

945 For the tensor flux we obtain

$$
\begin{aligned}
\mathcal{F}_{\mathcal{T}} = {} & \frac{8}{15} \frac{\eta^2}{G\alpha^2} \left(\frac{G\alpha M}{r}\right)^4 \Bigg\{ (12v^2 - 11\dot{r}^2) \\
& + \frac{1}{28}\Bigg[ -16\left( 170 - 10\eta + 63\bar{\gamma} + 84\beta_+ - 84\frac{\Delta M}{M}\beta_- \right)v^2 \frac{G\alpha M}{r} \\
& + (785 - 852\eta + 336\bar{\gamma})v^4 - 2(1487 - 1392\eta + 616\bar{\gamma})v^2\dot{r}^2 + 3\left( 687 - 620\eta + 280\bar{\gamma}\dot{r}^4 \right) \\
& + 8\left( 367 - 15\eta + 140\bar{\gamma} + 168\beta_+ - 168\frac{\Delta M}{M}\beta_- \right)\dot{r}^2 \frac{G\alpha M}{r} + 16(1 - 4\eta)\left(\frac{G\alpha M}{r}\right)^2 \Bigg] \Bigg\} \\
& + \frac{48G^3 M\alpha\mu}{5r^8}\left( 100\dot{r}^4 - 105\dot{r}^2 v^2 + 15v^4 + 18\frac{GM\alpha}{r}\dot{r}^2 - 11\frac{GM\alpha}{r}v^2 \right)\tilde{\zeta}_2 - \frac{64G^4 M^2\alpha^3\mu^2}{15r^7}(7\dot{r}^2 - 6v^2)\zeta_1 \\
& - \frac{192G^4 M^2\alpha^3\mu^2}{5r^9}(4\dot{r}^2 - 3v^2)\zeta_2.
\end{aligned}
\tag{107}
$$

946 Specializing to circular orbits and expressing the results in terms of $x$ leads to

$$
\mathcal{F}_T(x) = x^5 \left[ T5 + x\left( T6 + T6_{\text{tid},1}x^2 + T6_{\text{tid},2}x^4 \right) \right],
\tag{108}
$$

947 with the coefficients given by

$$
\begin{aligned}
T5 &= \frac{32\eta^2}{5\alpha^2 G}, \\
T6 &= \left(\frac{2\eta^2}{105\alpha^2 G}\right)\left( -1247 - 448\bar{\gamma} + 896\frac{\Delta M}{M}\beta_- - 896\beta_+ - 980\eta \right), \\
T6_{\text{tid},1} &= \frac{256\mu^2\zeta_1}{15\alpha^4 G^3 M^5}, \\
T6_{\text{tid},2} &= \left(\frac{192\mu\tilde{\zeta}_2}{5\alpha^7 G^5 M^7} + \frac{384\mu^2\zeta_2}{5\alpha^6 G^5 M^7}\right).
\end{aligned}
\tag{109}
$$

### B.0.3  GW phase evolution

949 In Sec. 4.5 we discuss the derivation of the DD and QD Fourier phase as a function of frequency.
950 In the DD domain the ratio obtained from energy balance in (70) is expanded as (72) with
951 coefficients

$$
\begin{aligned}
E_0' &= -\frac{3}{2} - \frac{\eta}{6} - \frac{4\bar{\gamma}}{3} + \frac{4}{3}\left(\beta_+ - \frac{\Delta M}{M}\beta_-\right) - x^2\frac{20\zeta_1}{3G^2\alpha^2 M^3} - x^4\frac{54\zeta_2}{G^4\alpha^4 M^5}, \\
f_2^{DD} &= \frac{24}{5\alpha S_-^2} + \frac{4S_+^2}{5S_-^2} - \frac{4\beta_+}{3} + \frac{4\beta_-\Delta M}{3M} - \frac{14}{5} - \frac{4\eta}{3} - \frac{2\bar{\gamma}}{3} + \frac{4\beta_-S_+}{\bar{\gamma}S_-} - \frac{4\beta_+\Delta M S_+}{\bar{\gamma}M S_-} + \frac{4\beta_+}{\bar{\gamma}} - \frac{4\beta_-\Delta M}{\bar{\gamma}M} \\
& + x^2\left( -\frac{\bar{\zeta}_1}{G^2 M^3\alpha^{7/2}S_-\mu} + \frac{4\zeta_1}{3G^2\alpha^2 M^3} \right) + x^4\frac{6\zeta_2}{G^4\alpha^4 M^5}.
\end{aligned}
\tag{110}
$$

952 The total DD phase within our approximations is given by

$$
\psi_{DD} = \frac{1}{4\eta S_-^2 x^{3/2}}\left\{ 1 + \rho^{DD}x + x^3\left[ \rho_{\text{tid}}^{DD}\log(x) - \frac{2}{3}\rho_{\text{tid}}^{DD} \right] - \frac{270\zeta_2}{7\alpha^4 G^4 M^5}x^5 \right\} + \phi_c - 2\pi f t_c,
\tag{111}
$$

953 with

$$
\rho^{DD} = -\frac{108}{5\alpha S_-^2} + 12\left(\beta_+ - \frac{\Delta M}{M}\beta_-\right) - \frac{9}{4\alpha^{3/2}M S_-}(\Delta M q_B\beta_A - M q_B\beta_A + \Delta M q_A\beta_B + M q_A\beta_B)
\tag{112}
$$

$$+ \frac{18}{\bar{\gamma}} \frac{S_+}{S_-} \left( \frac{\Delta M}{M} \beta_+ - \beta_- \right) - 3\bar{\gamma} + \frac{21\eta}{4} - \frac{18S_+^2}{5S_-^2} + \frac{117}{20}, \tag{113}$$

$$\rho_{\text{tid}}^{\text{DD}} = \frac{3}{G^2 \alpha^2 M^3 \eta} \left[ -\frac{\bar{\zeta}_1}{\alpha^{3/2} M S_-} + 16\eta \zeta_1 \right]. \tag{114}$$

954    In the QD domain, the energy balance ratio is expanded as (79) with coefficients given by

$$f_2^{nd} = \frac{1}{\xi} \left[ -\frac{8\beta_+}{3} - \frac{4\bar{\gamma}}{3} - \frac{35\eta}{12} + \frac{8\beta_- \Delta M}{3M} - \frac{1247}{336} + x^2 \frac{8\zeta_1}{3G^2 \alpha^2 M^3} + x^4 \left( \frac{6\tilde{\zeta}_2}{G^4 \alpha^5 M^5} + \frac{12\zeta_2}{G^4 \alpha^4 M^5} \right) \right],$$

$$f^d = -\frac{4\beta_+}{3} + \frac{4\beta_- \Delta M}{3M} - \frac{14}{5} - \frac{4\eta}{3} - \frac{2\bar{\gamma}}{3} + \frac{4\beta_- S_+}{\bar{\gamma} S_-} - \frac{4\beta_+ \Delta M S_+}{\bar{\gamma} M S_-} + \frac{4\beta_+}{\bar{\gamma}} - \frac{4\beta_- \Delta M}{\bar{\gamma} M}$$

$$+ x^2 \left( -\frac{\bar{\zeta}_1}{G^2 \alpha^{7/2} M^3 S_- \mu} + \frac{4\zeta_1}{3G^2 \alpha^2 M^3} \right) + x^4 \frac{6\zeta_2}{G^4 \alpha^4 M^5}. \tag{115}$$

955    The quadrupolar driven phase evolution is given by

$$\psi^{\text{QD}} = \psi_{\text{non-dip}} + \psi_{\text{dip}} + \phi_c + 2\pi f t_c, \tag{116}$$

956    with

$$\psi_{\text{non-dip}} = \frac{3\alpha}{128\eta \xi x^{5/2}} \left[ 1 + \rho^{nd} x + \rho_{\text{tid}}^{nd,1} x^3 + \rho_{\text{tid}}^{nd,2} x^5 \right], \tag{117a}$$

$$\psi_{\text{dip}} = -\frac{5S_-^2 \alpha^2}{1792\eta \xi^2 x^{7/2}} \left[ 1 + \rho^d x + \rho_{\text{tid}}^{d,1} x^3 \right.$$

$$\left. + \rho_{\text{tid}}^{d,2} x^5 \log(x) - \frac{2}{3} \rho_{\text{tid}}^{d,2} x^5 \right], \tag{117b}$$

957    and $\xi = 1 + S_+^2 \alpha / 6$. The non-dipolar coefficients are

$$\rho^{nd} = \frac{6235}{756\xi} - \frac{10}{3} - \left( 10 - \frac{175}{\xi} \right) \frac{\eta}{27} + \frac{80}{27} \left( \frac{1}{\xi} - 1 \right) \bar{\gamma}$$

$$+ \left( \frac{80}{27} + \frac{160}{27\xi} \right) \left( \beta_+ - \frac{\Delta M}{M} \beta_- \right), \tag{118a}$$

$$\rho_{\text{tid}}^{nd,1} = \frac{400\zeta_1}{3\alpha^2 G^2 M^3} + \frac{160\zeta_1}{3\alpha^2 G^2 M^3 \xi}, \tag{118b}$$

$$\rho_{\text{tid}}^{nd,2} = -\frac{24\tilde{\zeta}_2}{\alpha^5 G^4 M^6 \xi \eta} - \frac{216\zeta_2}{\alpha^4 G^4 M^5} - \frac{48\zeta_2}{\alpha^4 G^4 M^5 \xi}, \tag{118c}$$

958    and the dipolar parts are

$$\rho^d = \frac{1247}{96\xi} - \frac{301}{40} + \left( \frac{245}{24\xi} - \frac{21}{8} \right) \eta + \left( \frac{14}{3\xi} - \frac{7}{2} \right) \bar{\gamma}$$

$$+ \frac{7S_+}{\bar{\gamma} S_-} \left( \beta_- - \frac{\Delta M}{M} \beta_+ \right)$$

$$+ \left( \frac{28}{3\xi} + \frac{7}{\bar{\gamma}} \right) (\beta_+ - \frac{\Delta M}{M} \beta_-), \tag{119a}$$

$$\rho_{\text{tid}}^{d,1} = -\frac{35\bar{\zeta}_1}{2\alpha^{7/2} G^2 M^4 S_- \eta} - \frac{280\zeta_1}{3\alpha^2 G^2 M^3}$$

$$- \frac{280\zeta_1}{3\alpha^2 G^2 M^3 \xi}, \tag{119b}$$

$$\rho_{\text{tid}}^{d,2} = -\frac{140\tilde{\zeta}_2}{\alpha^5 G^4 M^6 \eta \xi} - \frac{560\zeta_2}{\alpha^4 G^4 M^5} - \frac{280\zeta_2}{\alpha^4 G^4 M^5 \xi}. \tag{119c}$$

## C Ready-to-use expressions for the tidal coefficients in the phase

For practical purposes, it is useful to study the combinations of tidal deformabilities and masses that appear in the GW phase with different powers of $x$. Here, we give explicit expressions for these terms.

### C.1 Quadrupolar-driven regime

The general structure of the tidal contribution to the phase in the QD regime was given in (82). Here, we provide the coefficients explicitly in terms of the NS properties.

The coefficient $c_2$ arising from dipolar tidal terms contributing to the dipolar piece of the phase read

$$c_2 = \frac{25}{3\alpha^2 G^2 M^3 \xi^3} \left\{ \left[ \frac{4}{3}(\xi+1)M_B q_B S_- + \frac{1}{4}\sqrt{\alpha}\xi M \right] q_B \frac{\lambda_A^{S,\ell=1}}{M_A} + \left[ \frac{4}{3}(\xi+1)M_A q_A S_- - \frac{1}{4}\sqrt{\alpha}\xi M \right] q_A \frac{\lambda_B^{S,\ell=1}}{M_B} \right\},$$
(120)

where the parameters $\alpha$, and $\xi$ were defined in (30), and (77) with (61), respectively. As this parameter (120) is multiplied by $S_-$ in the phase (82), in case of identical NSs where $S_- = 0$ it does not contribute.

Next, the parameter $c_3$ arising from the contribution of scalar dipolar tidal effects to the non-dipolar part of the Fourier phase is given by

$$c_3 = \frac{140}{3\alpha^3 \xi^2} \left[ \underbrace{\frac{4}{G^2 M^3} \left( M_B q_B^2 \frac{\lambda_A^{S,\ell=1}}{M_A} + M_A q_A^2 \frac{\lambda_B^{S,\ell=1}}{M_B} \right)}_{\to q^2 \Lambda_S^{\ell=1} \text{ for } M_A = M_B} + \frac{5}{42} \underbrace{\frac{(q_A+q_B)^2}{G^2 M^3} \left( M_B q_B^2 \frac{\lambda_A^{S,\ell=1}}{M_A} + M_A q_A^2 \frac{\lambda_B^{S,\ell=1}}{M_B} \right)}_{\to q^4 \Lambda_S^{\ell=1} \text{ for } M_A = M_B} \right],$$
(121)

and its value for identical NSs was given in (84).

Effects that involve all three Love numbers first appear in the quadrupolar tidal contributions to the dipolar piece of the phase, through the coefficient

$$\begin{aligned} c_4 = &\frac{50}{3\alpha^4 G^4 M^5 \xi^3} \left\{ [2(2\xi+1)M_B + \alpha M] \frac{\lambda_A^T}{M_A} + [2(2\xi+1)M_A + \alpha M] \frac{\lambda_B^T}{M_B} \right\} \\ &- \frac{50}{3\alpha^4 G^4 M^5 \xi^3} \left\{ [4(2\xi+1)M_B + \alpha M] \frac{\lambda_A^{ST}}{M_A} + [4(2\xi+1)M_A + \alpha M] \frac{\lambda_B^{ST}}{M_B} \right\} \\ &+ \frac{100(2\xi+1)}{3\alpha^4 G^4 M^5 \xi^3} \left( M_B q_B^2 \frac{\lambda_A^{S,\ell=2}}{M_A} + M_A q_A^2 \frac{\lambda_B^{S,\ell=2}}{M_B} \right), \end{aligned}$$
(122)

where $\lambda^T$ and $\lambda^{ST}$ are understood to be the quadrupolar $\ell = 2$ results, which is their lowest nontrivial multipolar order. Similarly to $c_2$, the coefficient (122) also appears multiplied by $S_-$ in the phase (82) and thus does not contribute in the equal-mass limit.

The coefficient of $x^5$ in (82) likewise has contributions from all three kinds of tidal deformabilities and is given by

$$\begin{aligned} c_5 = &\frac{39}{2\alpha^5 \xi^2} \left( \frac{7}{26} \underbrace{\frac{32(q_A+q_B)^2}{7G^4 M^5} \left[ \left( \frac{3M_B}{8} + \eta_q M \right) \frac{\lambda_A^T}{M_A} + \left( \frac{3M_A}{8} + \eta_q M \right) \frac{\lambda_B^T}{M_B} \right]}_{\to q^2 \Lambda_T^{\ell=2} \text{ for } M_A = M_B} \right. \\ &+ \frac{16}{26 G^4 (M\alpha)^5 \xi^2} \left\{ [2(9\xi+2)M_B + \alpha M] q_B \frac{\lambda_A^{ST}}{M_A} + [2(9\xi+2)M_A + \alpha M] q_A \frac{\lambda_B^{ST}}{M_B} \right\} \end{aligned}$$

$$- \frac{16(9\xi + 2)}{13G^4(M\alpha)^5\xi^2} \left[ q_B M_B \frac{\lambda_A^S}{M_A} + q_A M_A \frac{\lambda_B^S}{M_B} \right] \bigg),$$  (123)

where again $\lambda^T$, $\lambda^{ST}$ and $\lambda^S$ are understood to be quadrupolar $\ell = 2$. We have defined the symmetric charge ratio

$$\eta_q = \frac{q_A q_B}{(q_A + q_B)^2}.$$  (124)

For two identical bodies, the result (123) reduces to (85).

## C.2 Dipolar-driven regime

Similarly to the quadrupolar-driven phase, the dipolar-driven phase can be rewritten as

$$\psi_{DD} = \frac{1}{4\eta x^{3/2}} \left\{ (\text{pp}) + c_3^{DD} \left( \log x - \frac{2}{3} \right) x^3 + c_5^{DD} x^5 \right\},$$  (125)

with $c_3^{DD}$ and $c_5^{DD}$ given by

$$c_3^{DD} = -\frac{1}{G^2 M^3 \alpha^4 S_-^3} \left[ \left( 8M_B q_B - \sqrt{\alpha} M \right) q_B \frac{\lambda_A^{S,\ell=1}}{M_A} \right.$$
$$\left. + \left( 8M_A q_A + \sqrt{\alpha} M \right) q_A \frac{\lambda_B^{S,\ell=1}}{M_B} \right],$$  (126)

$$c_5^{DD} = -\frac{90}{7\alpha^6 G^4 M^5 S_-^2} \left[ \frac{M_B}{M_A} \left( q_B^2 \lambda_A^{S,\ell=2} - 2q_B \lambda_A^{ST,\ell=2} + \right. \right.$$
$$\left. \left. \lambda_A^{T,\ell=2} \right) + \frac{M_A}{M_B} \left( q_A^2 \lambda_B^{S,\ell=2} - 2q_A \lambda_B^{ST,\ell=2} + \lambda_A^{T,\ell=2} \right) \right].$$  (127)

For a system of two identical NSs, the scalar flux vanishes and there is no dipolar-driven phase.

# D  Gravitational waves in Jordan frame from Einstein frame results

In this appendix, we provide details on the relation of GWs in the Jordan frame to those in the Einstein frame that was briefly stated in Sec. 4.3. While this topic has already been discussed in the literature, e.g. [63,82,83], we generalize the results to arbitrary coupling functions and, in Sec. D.2 a result valid beyond the short-wave limit for describing GWs.

## D.1  Derivation based on the short-wave approximation

### D.1.1  Geodesic deviation and linearized frame transformations

In analyzing the basic physics of GW detection, it is useful to consider the deviation between geodesics of nearby test masses due to curvature induced by GWs. Our aim is to use this together with the frame transformations (2) to relate the Einstein-frame metric perturbations to physical effects on test masses in the Jordan frame.

In brief (see e.g. the review [97] for more details and [82] for the application to ST theories), we assume that one geodesic is at $z^\alpha(\tau)$, where $\tau$ is proper time, and a nearby one is at $z^\alpha(\tau) + L^\alpha(\tau)$, where $L^\alpha$ is small. The geodesic deviation in the Jordan frame is given by

$$\frac{D_*^2}{d\tau^2} L^\mu = R^\mu_{\alpha\nu\beta*} u^\alpha u^\beta L^\nu,$$  (128)

where $D_*/d\tau$ denotes the total derivative along the worldline. In the local proper detector frame, this reduces to

$$\ddot{L}^i = -R^i_{*0j0}L^j = \frac{1}{2}\ddot{h}^*_{ij}L^j,\tag{129}$$

with $h^*_{ij}$ the spatial parts of a small metric perturbation around a background metric $\eta^*_{\mu\nu}$

$$g^*_{\mu\nu} = \eta^*_{\mu\nu} + h^*_{\mu\nu}, \qquad g_{\mu\nu} = \eta_{\mu\nu} + h_{\mu\nu},\tag{130a}$$

where we have also written down the expansion of the metric in the Einstein frame. We recall that in the Einstein frame, vacuum gravity behaves as in GR and the spacetime of a compact binary source is asymptotically Minkowski, with $\eta_{\mu\nu}$ the standard Minkowski metric. Similarly to (130), we expand the scalar fields for small fluctuations around their asymptotic background values

$$\phi = \phi_0 + \delta\phi, \qquad |\delta\phi| \ll \phi_0,\tag{130b}$$

and likewise for the Einstein-frame field $\varphi$. Next, substituting (130) into the conformal transformation (2) and expanding to first order in the small quantities yields the relation [98]

$$h^*_{\mu\nu} = A^2_0\left(h_{\mu\nu} + 2\frac{A'_0}{A_0}\delta\varphi\,\eta_{\mu\nu}\right),\tag{131}$$

where the subscript 0 denotes evaluation of the function at $\varphi_0$. Using (131) to compute the Riemann tensor yields

$$R^k_{*0j0} = -\frac{1}{A^2_0}\eta^{ki}\frac{1}{2}\ddot{h}_{ij}.\tag{132}$$

Taking two time derivatives of (131) leads to

$$\ddot{h}^*_{ij} \simeq A^2_0\left(\ddot{h}_{ij} + \frac{2A'_0}{A_0}\delta\ddot{\varphi}\,\eta_{ij}\right),\tag{133}$$

at first order in the perturbations and at leading order in the distance to the source, i.e. we neglect terms such as $\dot{\varphi}_0\,\delta\varphi$. Substituting this into (132) yields

$$R^i_{*0j0} \simeq -\frac{1}{2}\left(\ddot{h}_{ij} + \frac{2A'_0}{A_0}\delta\ddot{\varphi}\,\eta_{ij}\right).\tag{134}$$

Comparing with (129), we can redefine the Jordan frame metric perturbation as

$$\ddot{h}^J_{ij} \equiv \frac{\ddot{h}^*_{ij}}{A^2_0} = \ddot{h}_{ij} + \frac{2A'_0}{A_0}\delta\ddot{\varphi}\,\eta_{ij}.\tag{135}$$

At first glance, one might think that there is a clash between (133) and (135) due to the factor of $A^2_0$. This is because (129) does not contain any information about the frame transformations. In particular, the quantity (135) is the relevant one for geodesic deviation in our approximations.

### D.1.2 Reduction to physical degrees of freedom

The next step is to fix a gauge in which only the physical degrees of freedom remain, such as the TT gauge in GR. We start by applying the transverse projector (64) on both sides of (135),

$$\ddot{h}^{J\,T}_{ij} \simeq \ddot{h}^T_{ij} + \frac{2A'_0}{A_0}\delta\ddot{\varphi}(\delta_{ij} - N_iN_j),\tag{136}$$

where $T$ stands for transverse, we have used that $\eta^{*T}_{ij} = P^k_i \eta^*_{km} P^M_j = P_{ij}$ and substituted (64). Taking the trace-less part yields

$$\ddot{h}^{J\,TT}_{ij} \simeq \ddot{h}^{TT}_{ij} \ . \tag{137}$$

This shows explicitly that the tensor plus and cross GW polarizations are the same in both frames. This also follows from expanding $\delta^\mu_\nu = g^\mu_\nu = g^{*\mu}_{\ \nu}$ with (130) to linear order in the perturbations, which likewise yields $h^i_j = h^{*i}_j$. As shown in [82], we can replace the term $\ddot{h}^{*\,T}_{ij}$ in (136) by $\ddot{h}^{*\,TT}_{ij}$. For the coupling $A(\varphi) = \varphi^{1/2}$ in (136), this reproduces the result in [63, 83, 99] with the redefinition $\Psi = \delta\varphi/\varphi_0$.

In summary, the Jordan-frame gravitational radiation can be computed from the Einstein-frame quantities via

$$h^{J\,T}_{ij} \simeq h^{TT}_{ij} + \frac{2A'_0}{A_0} \delta\varphi (\delta_{ij} - N_i N_j), \tag{138}$$

where for the setting considered here, $h^{TT}_{ij}$ is the waveform computed in Sec. 4.2 and given explicitly by (103), and $\delta\varphi$ the scalar waveform computed in Sec. 4.1 and given explicitly in (101).

### D.1.3 Polarization components

We next use the expansion (130) together with (4) to express the contribution of the Einstein frame scalar field in terms of the Jordan frame one,

$$-\frac{2A'_0}{A_0} \delta\ddot{\varphi} = \frac{F'}{F\sqrt{\Delta}}\bigg|_0 \delta\ddot{\varphi} \simeq \frac{F'(\phi_0)}{F(\phi_0)} \delta\ddot{\phi} \ . \tag{139}$$

Using this in the result for the metric perturbation (138), the decomposition into the fundamental polarization components discussed in [82] applies. This shows that in addition to the plus and cross tensorial polarizations, there is also a scalar mode. As discussed in [82], for theories with a coupling to the Ricci scalar of the form $F(\phi) = \phi$ in (1) and wave propagation in the $z$-direction

$$h^J_{ij}\,|_{F(\phi)=\phi} = h_+ e^{(+)}_{ij} + h_\times e^{(\times)}_{ij} - \frac{\delta\phi}{\phi_0} e^S_{ij} \ , \tag{140}$$

with the amplitude of the scalar polarization obtained by using (139) together with the identification of $\delta\varphi$ as the scalar radiation field in the Einstein frame computed in Sec. 4.1.

### D.1.4 Asymptotic flatness and size of the scalar GWs in the Jordan frame

From the expressions derived above, we can draw several interesting conclusions. The first is regarding the asymptotic background metric $\eta^*_{\mu\nu}$ that appears in (130). From the frame transformation (2), it follows that that $\eta^*_{\mu\nu} = A(\varphi_0)^2 \eta_{\mu\nu}$, with $\eta_{\mu\nu}$ the Minkowski metric. The transformation indicates that in general, the Jordan frame metric is not asymptotically flat. There are two ways to avoid this difficulty: one is to rescale the coordinates to absorb the factor $A^2_0$, such that the line element $ds^2_*$ in the Jordan frame is indeed the correct flat-space one [16, 45]; another way is to rescale the conformal factor such that $A(\varphi) \to A(\varphi)/A(\varphi_0)$, or equivalently imposing $A(\varphi_0) = 1$, for which no redefinition is necessary.

A second interesting point is that due to the scalar-field contribution, the metric perturbation in the Jordan frame is only transverse but not traceless. However, depending on the coupling function, the scalar contribution to the waveform (135) can be strongly suppressed, due to the factor of $A'(\varphi_0)/A(\varphi_0)$. For instance, for theories with generic power law couplings

1060   $A(\varphi) = \varphi^\kappa$, dilatonic couplings $A(\varphi) = e^{\gamma\varphi}$, or Gaussian couplings [20]$A(\varphi) = e^{\frac{1}{2}\beta_0\varphi^2}$, de-
1061   pending on the coupling constants and values of $\varphi_0 = \varphi_\infty$, the scalar field contribution to the
1062   Jordan-frame waveform may be suppressed by many orders of magnitude.

### D.2   Derivation beyond the short-wave approximation

1064   The results in Sec. D.1 above can be applied to ground-based detectors as long as the size of
1065   the arms $L$ is shorter than the reduced GW wavelength $\tilde{\lambda}$, $L \ll \tilde{\lambda}$. To study GWs beyond this
1066   approximation requires analyzing the scattering of light between mirrors as we now discuss.
1067   Similarly to (128), the time it takes for the light of the interferometer to scatter from the mirror
1068   and come back, is given by the same expression as in GR, with their corresponding metrics.
1069   We will proceed as above and start from the physical Jordan frame, and at the end rewrite
1070   the expressions in terms of Einstein frame quantities. Similarly to [100], we consider the plus
1071   polarization of a gravitational wave moving along the $z$-axis. We will focus on two different
1072   kinds of orientations for the interferometer arms, when they lie along the $x - y$ or the $x - z$
1073   plane.

1074        In this set-up, we have

$$
\begin{aligned}
ds^2 = &-dt^2 + [1 + h_+ + \Phi]dx^2 \\
&+ [1 - h_+ + \Phi]dy^2 + dz^2 \,,
\end{aligned}
\tag{141}
$$

1075   with $\Phi$ defined in (140). For null rays, $ds^2 = 0$ and, at first order in $h_+$ and $\Phi$, for the $x-$arm,

$$
dx = \pm \left\{ 1 - \frac{1}{2}[h_+ + \Phi] \right\} dt \,.
\tag{142}
$$

1076   For a photon travelling from the origin $(x, t) = (0, t_0)$ to the mirror at $(x, t) = (L_x, t_1)$, we
1077   integrate (142) with the plus sign. For the return trip, from $(x, t) = (L_x, t_1)$ to $(x, t) = (0, t_2)$,
1078   we use the minus sign.

1079        The time it takes for a photon to complete $N$ round trips along the $x$-arm is given, at first
1080   order in the perturbations, by

$$
\begin{aligned}
\Delta t^{(x)} &= N(t_2 - t_0) \\
&= 2NL_x + \frac{N}{2} \int_{t_0}^{t_0 + 2L_z} \left[ h_+(t') + \Phi(t') \right] dt' \,,
\end{aligned}
\tag{143}
$$

1081   where we have substituted $t_2 = t_0 + 2L_x + \mathcal{O}(h_+, \Phi)$ in the upper limit of the integral. Consid-
1082   ering a plane wave as a simple example,

$$
h_+ = A^+ \cos(\omega_{\text{gw}} t) \,, \qquad \Phi = \Phi_0 \cos(\omega_s t) \,,
\tag{144}
$$

1083   with $\omega_{\text{gw}}$ and $\omega_s$ the frequency of the GW and the scalar field, respectively, and using

$$
\int_{t_0}^{t_0 + 2L_x} \cos(\omega t) = \frac{\sin(\omega L_x)}{(\omega L_x)} \cos[\omega(t_0 + L_x)] \,,
\tag{145}
$$

1084   yields,

$$
\begin{aligned}
\Delta t^{(x)} = &2NL_x \\
&\left\{ 1 + \frac{1}{2}\left[ h_+(t_0 + L_x) + \Phi(t_0 + L_x)\frac{\omega_{\text{gw}}}{\omega_s}\frac{\sin(\omega_s L_x)}{\sin(\omega_{\text{gw}} L_x)} \right] \right.
\end{aligned}
$$

$$\times \frac{\sin(\omega_{gw}L_x)}{(\omega_{gw}L_x)} \Bigg\} \;, \tag{146}$$

or, considering that we observe the wave at some given time $t \equiv t_2 = t_0 + 2L_x$, yields

$$\Delta t^{(x)} = 2NL_x$$
$$\times \left\{ 1 + \frac{sc_{gw,x}}{2} \left[ h_+(t - L_x) + \Phi(t - L_x)\frac{sc_{s,x}}{sc_{gw,x}} \right] \right\} \;, \tag{147}$$

where we define

$$sc_{p,q} \equiv (\omega_p L_q) = \frac{\sin(\omega_p L_q)}{(\omega_p L_q)} \;. \tag{148}$$

For the $y$-arm the result is similar but with $h_+ \to -h_+$. The phase-shift $\Delta\Theta$ produced in the interferometer is given by

$$\Delta\Theta^{(x-y)} = \omega_L \left( \Delta t^{(x)} - \Delta t^{(y)} \right)$$
$$= 2N\omega_L \Delta L$$
$$\left\{ 1 + \frac{L^+}{2\Delta L} \left[ h_+(t_+)sc_{gw,+} + \Phi(t_+)sc_{s,+} \right] \right.$$
$$\left. + \frac{L^-}{2\Delta L} \left[ h_+(t_-)sc_{gw,-} - \Phi(t_-)sc_{s,-} \right] \right\} \;, \tag{149}$$

with $\omega_L$ the laser frequency, and the definitions

$$L \equiv \frac{L_x + L_y}{2} \;, \quad \Delta L \equiv L_x - L_y \;,$$
$$L^{+/-} \equiv L \pm \frac{\Delta L}{2} \;, \quad t_{+/-} = t - L^{\pm} \;. \tag{150}$$

For an interferometer with equal-length arms, $\Delta L = 0$, $L^+ = L^- = L$ and

$$\Delta\Theta^{(x-y)} = h_+(t - L)\Theta_{arm}\frac{\sin(\omega_{gw}L)}{(\omega_{gw}L)} \;, \tag{151}$$

with $\Theta_{arm} = 2LN\omega_L$ . This is the same result as in GR. This is because the scalar field contributes equally in the $x$ and $y$ directions, and therefore its oscillations will cancel out.

We now focus on the case where the arms are in the $x$-$z$ axis. In this case, the expression for the $x$-arm is the same as above, and for the $z$-axis we have

$$\Delta t^{(z)} = 2NL_z \;, \tag{152}$$

since the wave is propagating along the $z$ axis and is transverse. In this case, there is no contribution of the scalar field along the $z$-axis and the phase yields

$$\Delta\Theta^{(x-z)} = \omega_L \left( \Delta t^{(x)} - \Delta t^{(z)} \right)$$
$$= 2N\omega_L \Delta L$$
$$\left\{ 1 + \frac{L^+}{2\Delta L} \left[ h_+(t_+)sc_{gw,+} + \Phi(t_+)sc_{s,+} \right] \right\} \;, \tag{153}$$

with the change of label $y \rightarrow z$ in the definitions (150). For the case that the arms have equal length, we obtain

$$\Delta\Theta^{(x-z)} = \frac{\Theta_{\text{arm}}}{2} \left[ h_+(t-L)sc_{\text{gw},+} + \Phi(t-L)sc_{s,+} \right] . \tag{154}$$

Using (138), in terms of the Einstein frame waveforms we have

$$\Delta\Theta^{(x-y)} = \Theta_{\text{arm}} h_+^*(t-L)sc_{\text{gw},+} , \tag{155}$$

$$\Delta\Theta^{(x-z)} = \frac{\Theta_{\text{arm}}}{2} \left[ h_+^*(t-L)sc_{\text{gw},+} - 2\alpha_\infty \delta\varphi(t-L)sc_{s,+} \right] . \tag{156}$$

Note that the lengths and frequencies will be the same as long as we are far from the scalar field. In the short-wave approximation, $sc \rightarrow 1$ and we recover the result in [82] for the scalar field contribution.

# E  Elaboration on the parameter space study

In Sec. 5.4.2 we did a phase space analysis of the tidal contributions to the QD Fourier phase captured by the $c_i$ coefficients in (82). Based on these parameter space studies we selected three types of systems that maximize the different types of tidal contributions and an intermediate scenario both for NS-NS and BH-NS systems. In this section in the main text we showed two of these parameter space plot explicitly in Fig. 7 for the sake of readability. In this appendix we elaborate on this analysis of Sec. 5.4.2, showing the parameter space results for the other coefficients in (82) in Fig. 10. In Fig 10 we show the contour plots for the different $c_i$ coefficients defined in (82), in the $(M_A, M_B)$ parameter space. We fix the SLy EoS, although the discussion is qualitatively similar for the WFF1 and H4 EoS.

The contour plots for the quadrupolar tensor tidal contributions captured in $(\frac{39}{2\alpha^5\xi^2}\tilde{\Lambda} + c_5)$ are for NS-NS and BH-NS systems qualitatively similar, in that the maximum absolute value is reached for the lowest masses. This is because the coefficients depend on the inverse of the total mass, which is maximum towards the limit of the lower NS mass of $1M_\odot$. Hence, our first choice of types of systems that would be interesting to study with respect to maximizing the quadrupolar tensor tidal contributions is that of the lowest possible mass for both NS-NS and BH-NS systems

$$(M_A, M_B)_1^{\text{NS}-\text{NS}} = (1M_\odot, 1M_\odot) , \tag{157a}$$

$$(M_A, M_B)_1^{\text{BH}-\text{NS}} = (5M_\odot, 1M_\odot) . \tag{157b}$$

The absolute value of the other contributions in (82) for both systems is peaked when one of the masses corresponds to that of the maximum charge $M_q \equiv M(q_{\text{max}})$, and the other mass corresponds to the lowest possible configuration. This can be interpreted as scalar-tensor effects being more noticeable as the charge is maximum, which indeed characterises the strength of the scalar field. Therefore, our second choice of systems will be

$$(M_A, M_B)_2^{\text{NS}-\text{NS}} = (1M_\odot, M_q) , \tag{158a}$$

$$(M_A, M_B)_2^{\text{BH}-\text{NS}} = (5M_\odot, M_q) . \tag{158b}$$

Additionally, for a NS-NS system, the $c_3$ contribution reaches moderate values when $M_A = M_B = M_q$. This can be explained with that this contribution contains the dipolar scalar tidal contribution in $\zeta_1$, which is maximum for and equal-mass system at the maximum charge(see Fig. 3). Additionally, the prefactors in front of the tidal deformabilities in the $c_i$ contributions in (82) also

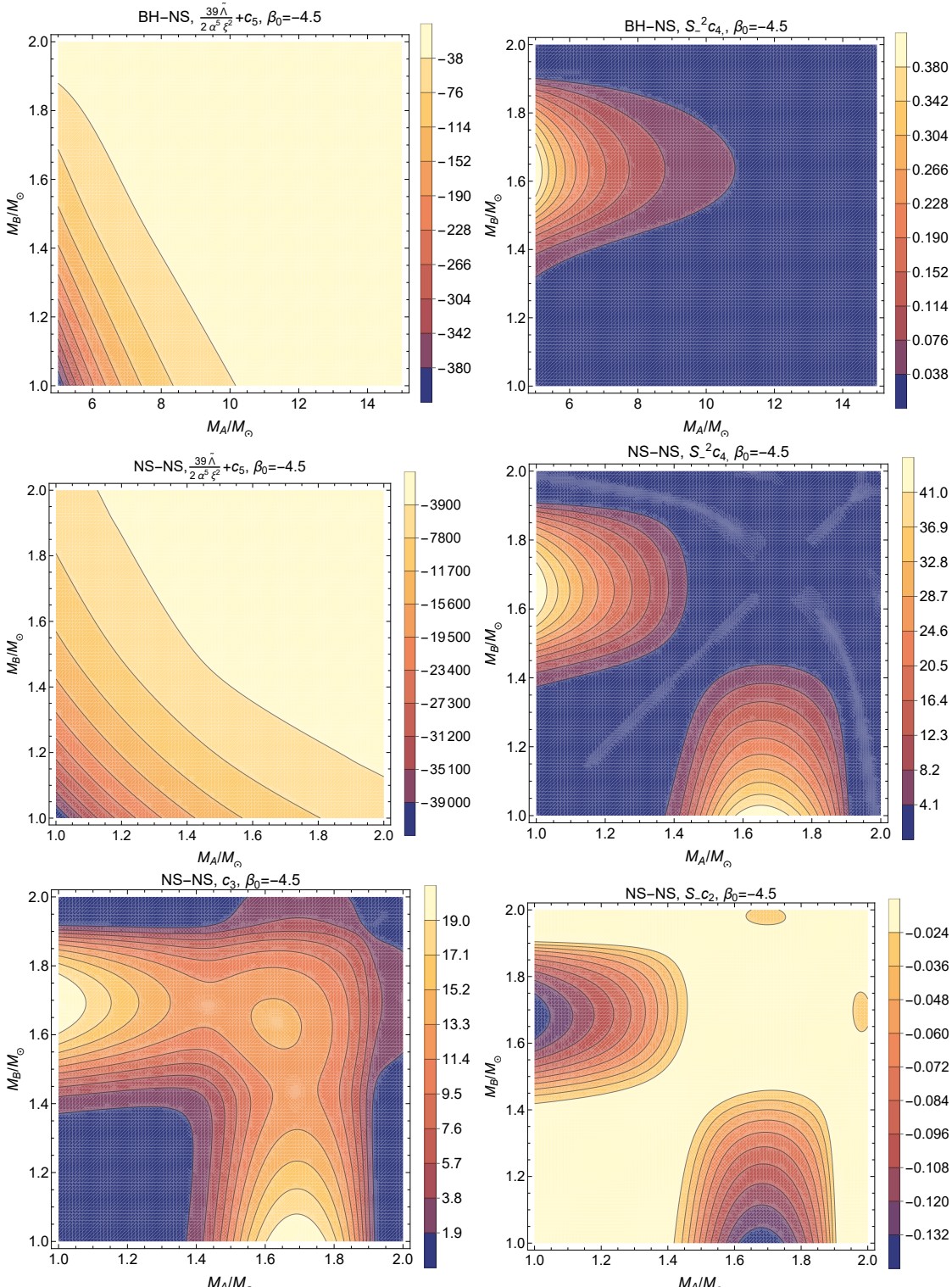

Figure 10: Contour plots of the $c_i$ coefficients defined in (82), in the $M_A - M_B$ parameter space for $\beta_0 = -4.5$ and the SLy EoS for a BH-NS system (top) and NS-NS system (middle and bottom). The plots for the other equations of state are qualitatively similar.

depend on the scalar charges and are maximised for the equal-mass, maximum-charge configurations, contributing to an overall lower effect than when considering one of the companions with the lowest mass, for which $q \sim 0$. Finally, we will also choose an intermediate case where effects are moderate,

$$(M_A, M_B)_3^{\text{NS−NS}} = (1.1 M_\odot, 1.2 M_\odot) \,, \tag{159a}$$

$$(M_A, M_B)_3^{\text{BH−NS}} = (8.9 M_\odot, 1.2 M_\odot) \,, \tag{159b}$$

where we choose slightly different masses for the NS-NS system to avoid the vanishing of the term $S_-$ in the phase.

In the main body of the paper we refer to (157a), (158a) and (159a) as scenarios 1, 2 and 3 respectively.

## F  Dipolar Driven phase evolution

As discussed in Sec. 5.4.1 the DD domain for the parameter space of NS-NS and BH-NS systems we studied lies below the lower bound frequency of the groundbased detectors. However it might still be relevant to study the DD domain as the boundary frequency is an approximation and the transfer frequency regime might be a broader range extending to higher frequencies [49]. In this appendix we show the results of the tidal contributions to the DD fourier phase (111) in Fig. 11. The dipolar driven regime is not present for equal mass systems for which $S_- = 0$, so only Scenario 2 and 3 are non-trivial. The contributions are shown for frequencies $10^{-5}$Hz to the boundary frequency (71). We choose to show here the different tidal contributions per Scenario instead of the contribution for the different systems; as in the DD regime the boundary frequency changes for different systems hence it is clearer to show the results per system. We find that the quadrupolar contributions are for both scenario's much smaller than in the QD regime. The dipolar scalar tidal contributions are clearly dominating which is expected in the DD domain where the dipolar radiation dominates over the quadrupole terms. The largerst dipolar tidal contribution is found in Scenario 2, becoming of order 1 near the boundary frequency.

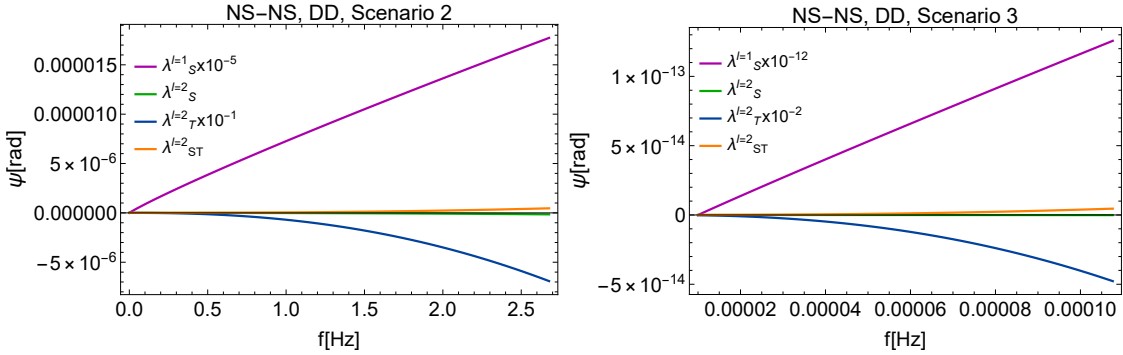

Figure 11: Different contributions of the tidal deformabilities to the Fourier phase for a NS-NS system in the DD regime for the scenario 2 and 3 in Sec. 2. Note that some of the contributions divided by orders of 10 to show the curves properly in one figure. The frequency domain is cut of at the boundary frequency of te DD domain (71).

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
