# Peer review of "Tidal effects in gravitational waves from neutron stars in scalar-tensor theories of gravity"

_SciPost Physics Core_

## Round 1 · Referee Report · Anonymous (Referee 1) · 2025-2-19

Report

This manuscript present a very detailed study on tidal effects in neutron-star binaries in an extension to General Relativity, namely scalar tensor theory. It goes all the way from computation of the (gravitational and scalar) tidal Love numbers down to their effect in the waveform emitted by a binary system within a post-Newtonian scheme.

Although the computation is involved, it is sound and most intermediate steps are given in order to reproduce it. This work pushes forward the state of the art for tidal effects beyond General Relativity and will be a reference for future studies.

For these reasons, I warmly recommend this high-quality work for publication, and I believe it meets all criteria for publication in SciPost Physics Core. I have two comments for the authors:

  • What's g^{LP} introduced in eq.17?

  • One of the main findings is that tidal effects enter at lower post-Newtonian order than in General Relativity (eg. the equation in the conclusion). The authors claim that, because of this fact and the different signs associated with different post-Newtonian tidal corrections in the GW phase, the net tidal signatures in scalar-tensor gravity are smaller than in General Relativity. I do not quite understand this conclusion. Naively, I would have expected that the net signatures should be larger, since they enter at lower post-Newtonian order, and the leading term should dominate. Why isn't this case? Is it because a different scaling of the scalar Love number (entering at x^2 in the equation of Sec. 6) with the compactness compared to the usual x^5 term? In case, might one expect that for very compact stars, where the compactness term (M/R)^5 is less important, eventually the scalar contribution at O(x^2) becomes dominant?

Recommendation

Publish (surpasses expectations and criteria for this Journal; among top 10%)

  • validity: top
  • significance: high
  • originality: high
  • clarity: top
  • formatting: perfect
  • grammar: perfect

Author:  Iris van Gemeren  on 2025-04-13  [id 5366]

(in reply to Report 1 on 2025-02-19)
Category:
answer to question

We would like to thank the referee for the evaluation, comments and recommendation. We respond to the comments below and incorporated these additions in the manuscript for resubmission.

• We added the definition of $g^{LP}$ in eq 17 as the product of metrics with the string of indices corresponding to the values of L and P
\begin{equation}
g^{L P}=\prod_{n=1}^{\ell} g^{l_n p_n}.
\end{equation}

• It is indeed correct that we found that the different types of tidal effects enter the phase evolution at effective lower PN order that the GR tidal term, tracked by the scaling with PN parameter x.

Intuitively, one would expect the lowest PN tidal contribution to dominate, in this case this would come from the scalar dipole tidal contribution, scaling with x^2 compared to the GR x^5 contribution as mentioned by the referee.

However the construction of the phase expression comes from the dubble independent expansions in PN and small tidal effects. Taking into account both the PN plus tidal order changes the hierarchy compared to only considering the PN orders. See also our discussion on this subtle issue in section IIIC of arxiv 2302.08480

More practically put, the coefficients in front of the PN factors of x in eq. 83 differ in orders of magnitude as shown in the figure 7, 10 and summarized in table 1. The magnitude of the coefficients have complex dependencies on the masses of the bodies, tidal deformability parameters, scalar charges and implicitly via these parameters also on the compactness of the bodies. The dependencies cannot be straightforwardly read of the expressions of the coefficients, therefore we did the parameter studies in the figures mentioned above.

We found (see fig. 9) for the parameter space of NS masses and BH masses we studied, that the tidal contributions coming from the quadrupolar tidal effects dominate for all studied systems, even though it comes with the highest PN order (x^5) in the phase evolution. Next in the hierarchy comes the contribution from the dipolar scalar tidal terms. As the scalar dipolar phase correction comes with opposite sign compared to the dominant quadrupolar tidal term, it decreases the overall magnitude of the tidal correction to the phase. Note however that this is the case in the QD frequency regime. For lower frequencies in the DD regime, the dependence on x becomes more important and the scalar dipolar tidal contribution are dominant which we showed in appendix F.
The difference in hierarchy compared to the one you would expect from the PN ordering is an important part of our conclusions and we added a greater emphasis on this in our discussion section.

---

## Round 1 · Referee Report · Anonymous (Referee 2) · 2025-3-18

Report

In the manuscript "Tidal effects in gravitational waves from neutron stars in scalar-tensor theories of gravity" the authors compute tidal corrections - up to the quadrupolar order - to gravitational-wave observables from coalescing binary systems composed of neutron stars in scalar-tensor theories. In the paper, after providing a quite comprehensive review on the basic formalism that will be used throughout the paper and justifying all the regimes of validity of their approach, they derive important formulae for the tidal contributions to the binding energy and both scalar and tensor waveforms, computed for arbitrary multipole moments, as well as for the energy fluxes, truncated at the quadrupolar level. These quantities are computed to first order in the PN expansion and in the small parameter that characterizes the scale of tidal deformability. After this, tidal effects on the phase evolution are studied up to the quadrupolar order. The discussion held until then is kept general, valid for any types of neutron stars in scalar-tensor theories. In the following sections, the authors apply the previous results to specific cases of astrophysical relevance, providing a detailed study on how tidal contributions affect the gravitational waves and on other relevant quantities in these cases.

The paper is very well structured and very well written, with abstract and introduction conveying and justifying with clarity its main goals. The paper presents a substantial amount of new results, all being safely scientifically sound and well justified. In particular, the analysis of the specific cases in the second part of the paper is of high quality, specially due to how detailed it is, always bringing discussions on the feasibility of detecting the studied tidal signatures using both present and future gravitational-wave observatories, highlighting the importance of their work. For all these reasons, I highly recommend the publication of this manuscript in SciPost Physics Core.

Below, I suggest small modifications, also pointing out a few typos I could identify:

  • Line 104: It seems there's something wrong with the label of Ref. [64]. Note that we see [54-64,64-67] instead of [54-67].

  • Eq. (17): Please inform what $g^{LP}$ is.

  • Line 267: Eq. (14) is cited when it seems to me that you should have cited Eq. (13) instead.

  • Is Eq. (32) really accurate? Since $x_A$ and $x_B$ are independent variables, we should have two Euler-Lagrange equations, one for body $A$ and one for body $B$.

  • In sections 4.1 and 4.2, it may not be clear to the reader why while the point-particle contribution of the tensor quadrupole (in Eq. (52)) contribute to the tensor waveform (in Eq. (56)), but the similar contribution for the scalar moments (in Eq. (42)) does not contribute to the scalar waveform (in Eq. (50)). A comment (or, maybe, a footnote) below Eq. (57) would be very much appreciated.

  • Lines 477-478: The first sentence of Sec. 4.5 is very confusing. Please, rewrite it.

  • Eq. (76): Please point to the reader where the coefficients $f_2^{nd}$ and $f_2^d$ can be explicitly found. (By the way, it seems you forgot the subscript "2" of $f_2^{d}$ in Eq. (115).)

  • In Fig. (5): Is there any particular reason why you exchanged the labels for the third plot? While pp and pp+tidal are represented by a solid blue line and a orange dashed line in the first two plots, respectively, this is changed in the third. If it's easy and quick to generated this plot, I would recommend standardizing this. Otherwise, this is a minor point.

Typos:

  • Lines 68-69: "a significant research efforts". Either remove "a" or efforts $\rightarrow$ effort.

  • Line 238: charactersitics $\rightarrow$ characteristics.

  • Line 364: Greens $\rightarrow$ Green's.

  • Line 674: Inverted quotation marks, 'unambiguos BH candidates'.

Recommendation

Ask for minor revision

  • validity: top
  • significance: high
  • originality: high
  • clarity: top
  • formatting: perfect
  • grammar: perfect

Author:  Iris van Gemeren  on 2025-04-13  [id 5367]

(in reply to Report 2 on 2025-03-18)
Category:
answer to question

Thank you for the evaluation, comments and recommendation. We respond to the comments below and explain how they are incorporated into the manuscript for resubmission.

- We corrected the label of citation [64]

- We added the definition of $g^{LP}$ in eq 17 as the product of metrics with the string of indices corresponding to the values of L and P
\begin{equation}
g^{L P}=\prod_{n=1}^{\ell} g^{l_n p_n}.
\end{equation}

- In our submitted version of the manuscript this should indeed have been eq. (14) instead of (13) as we considered both the point particle and tidal contributions to the action in the field equations. Then we discussed based on the results of arxiv 2302.08480 that the tidal contributions to the near zone fields were of higher PN order and our results then only captured the tidal contribution to the dynamics directly coming from the tidal action in sec. 3.2.
However a correction was pointed out to us for arxiv 2302.08480 featuring an improper treatment of the tidal contribution to the near zone fields, recovering that they are vanishing at 1PN which should not be the case.
Luckily both in arxiv 2302.08480 and this manuscript we did consider the contribution directly from the tidal action and as is shown in appendix A of arxiv 2310.19679 this yields the same results as letting the tidal corrections enter via the near zone fields at the PN order we are considering. Therefore our results do not change but we removed the consideration of the tidal contributions at the level of the field equations and only considered the tidal corrections at the level of the Lagrangian in sec. 3.2, referring to arxiv 2310.19679 to verify that these methods are indeed equivalent. Hence referring to eq (13) is now correct.

- It is correct that as xA and xB are independent variables there are two Euler-Lagrange equations for the two separate bodies. Now we are interested in the relative acceleration of the two bodies, therefore considering the difference between these two equations resulting in eq. 32.

- Thank you for pointing this out. The reason that this contribution is not present for the scalar waveform is that they come in at a higher PN order, we included a footnote at the suggested place explaining this.

- We rewrote this sentence to ‘Generally, the tidal corrections compared to the GR contributions in the radiation are small. However, during the many GW cycles in the inspiral these corrections accumulate in the phase evolution’.

- We added a reference to the expressions of the coefficients and corrected the subscript ‘2’.

- We changed the labelling of the bottom plot in Fig.5 to be consistent with the plots above.

- We corrected the typo’s as suggested by the referee.

---

## Editorial Decision

resubmitted